# Practical Conditional Neural Processes Via Tractable Dependent Predictions

**Stratis Markou***
University of Cambridge
em626@cam.ac.uk

**James Requeima***
University of Cambridge
Invenia Labs
jrr41@cam.ac.uk

**Wessel P. Bruinsma***
University of Cambridge
Invenia Labs
wpb23@cam.ac.uk

**Anna Vaughan**
University of Cambridge
av555@cam.ac.uk

**Richard E. Turner**
University of Cambridge
ret26@cam.ac.uk

## Abstract

Conditional Neural Processes (CNPs; Garnelo et al., 2018a) are meta-learning models which leverage the flexibility of deep learning to produce well-calibrated predictions and naturally handle off-the-grid and missing data. CNPs scale to large datasets and train with ease. Due to these features, CNPs appear well-suited to tasks from environmental sciences or healthcare. Unfortunately, CNPs do not produce correlated predictions, making them fundamentally inappropriate for many estimation and decision making tasks. Predicting heat waves or floods, for example, requires modelling dependencies in temperature or precipitation over time and space. Existing approaches which model output dependencies, such as Neural Processes (NPs; Garnelo et al., 2018b) or the FullConvGNP (Bruinsma et al., 2021), are either complicated to train or prohibitively expensive. What is needed is an approach which provides dependent predictions, but is simple to train and computationally tractable. In this work, we present a new class of Neural Process models that make correlated predictions and support exact maximum likelihood training that is simple and scalable. We extend the proposed models by using invertible output transformations, to capture non-Gaussian output distributions. Our models can be used in downstream estimation tasks which require dependent function samples. By accounting for output dependencies, our models show improved predictive performance on a range of experiments with synthetic and real data.

## 1 Introduction

Conditional Neural Processes (CNP; Garnelo et al., 2018a) are a scalable and flexible family of meta-learning models capable of producing well-calibrated uncertainty estimates. CNPs naturally handle off-the-grid and missing data and are trained using a simple-to-implement maximum-likelihood procedure. At test time, CNPs require significantly less computation and memory than other meta-learning approaches, such as gradient-based fine tuning (Finn et al., 2017; Triantafillou et al., 2019), making them ideal for resource and power-limited applications, such as mobile devices. Further, CNPs can be combined with attention (Kim et al., 2019), or equivariant networks which account for symmetries in the task at hand (Gordon et al., 2020; Kawano et al., 2021; Holderrieth et al., 2021), achieving impressive performance on a variety of problems. Despite these favourable qualities, CNPs are severely limited by the fact that they do not model dependencies in their output (fig. 1).

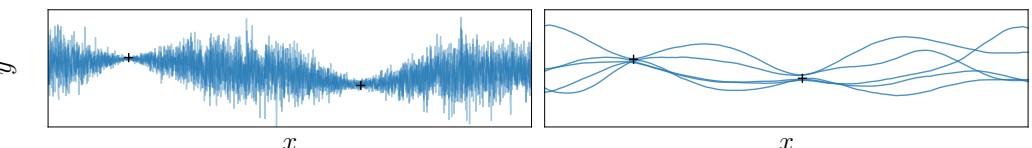

Figure 1: Unlike the ConvCNP (left) which makes independent predictions, the ConvGNP (right), introduced in this work, makes dependent predictions and can be used to draw function samples (blue) which are coherent. These are often necessary for downstream estimation tasks.

---

*Authors contributed equally.

**Limitations of CNPs:** More specifically, given two target input locations $x_m$ and $x_{m'}$, CNPs model their respective outputs $y_m$ and $y_{m'}$ independently. In this paper, we refer to such predictions as *mean-field*. The inability to model dependencies hurts the predictive performance of CNPs and renders it impossible to produce coherent function samples. Since many downstream tasks require dependent function samples, this excludes mean-field CNPs form a range of applications. In heatwave or flood prediction for example, we need to evaluate the probability of the event that the temperature or precipitation remains above some threshold, throughout a region of space and time. As illustrated by fig. 1, mean-field predictions model every location independently, and may assign unreasonably low probabilities to such events. If we were able to draw coherent samples from the predictive, the probabilities of such events and similar useful quantities could be more reasonably estimated.

**Limitations of existing models with dependencies:** To address the above, follow-up work has introduced Neural Processes (NPs; Garnelo et al., 2018b; Kim et al., 2019; Foong et al., 2020), which use latent variables to model output dependencies. However, the likelihood for these models is not analytically tractable, so approximate inference is required for training (Le et al., 2018; Foong et al., 2020). Alternatively, Bruinsma et al. (2021) recently introduced a variant of the CNP called the Gaussian Neural Process, which we will refer to as the FullConvGNP, which directly parametrises the covariance of a Gaussian predictive over the output variables. In this way the FullConvGNP models statistical dependencies in the output, and can be trained by an exact maximum-likelihood objective, without requiring approximations. However, for $D$-dimensional data, the architecture of the FullConvGNP involves $2D$-dimensional convolutions, which can be very costly, and, for $D > 1$, poorly supported by most Deep Learning libraries.

**Contributions:** In this work (i) we introduce Gaussian Neural Processes (GNPs), a class of model which directly parametrises the covariance of a Gaussian predictive process, thereby circumventing the costly convolutions of the FullConvGNP, and is applicable to higher-dimensional input data. GNPs have analytic likelihoods making them substantially easier to train than their latent variable counterparts; (ii) we show that GNPs can be easily applied to multi-output regression, as well as composed with invertible marginal transformations to model non-Gaussian data; (iii) we demonstrate that modelling correlations improves performance on experiments with both Gaussian and non-Gaussian synthetic data, including a downstream estimation task that mean-field models cannot solve; (iv) we demonstrate that GNPs outperform their mean-field and latent variable counterparts on real-world electroencephalogram (EEG) data and climate data; (v) in climate modelling, GNPs outperform a standard ensemble of widely used methods in statistical downscaling, while providing spatially coherent temperature samples which are necessary for climate impact studies.

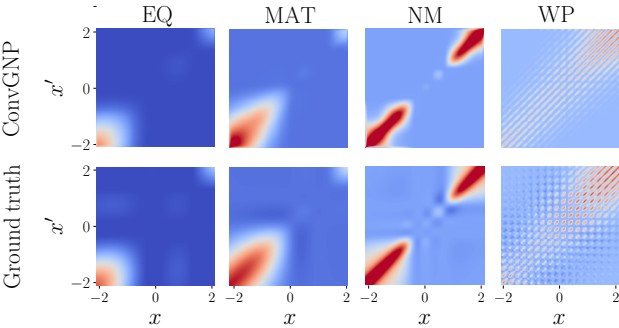

Figure 2: The ConvGNP model, introduced in this work, can recover intricate predictive covariances. Columns show the posterior covariances produced by a ConvGNP, after training with synthetic data drawn from a Gaussian Process with a different covariance (exponentiated quadratic, Matern, noisy mixture or weakly periodic), and conditioned on a randomly sampled dataset.

## 2   CONDITIONAL & GAUSSIAN NEURAL PROCESSES

**Background:** We present CNPs from the viewpoint of *prediction maps* (Foong et al., 2020). A prediction map $\pi$ is a function which maps (1) a *context set* $(\mathbf{x}_c, \mathbf{y}_c)$ where $\mathbf{x}_c = (x_{c,1}, \dots, x_{c,N})$ are the inputs and $\mathbf{y}_c = (y_{c,1}, \dots, y_{c,N})$ the outputs and (2) a set of *target inputs* $\mathbf{x}_t = (x_{t,1}, ..., x_{t,M})$ to a distribution over the corresponding *target outputs* $\mathbf{y}_t = (y_{t,1}, ..., y_{t,M})$:

$$\pi(\mathbf{y}_t; \mathbf{x}_c, \mathbf{y}_c, \mathbf{x}_t) = p(\mathbf{y}_t | \mathbf{r}), \tag{1}$$

where $\mathbf{r} = r(\mathbf{x}_c, \mathbf{y}_c, \mathbf{x}_t)$ is a vector which parameterises the distribution over $\mathbf{y}_t$. For a fixed context $(\mathbf{x}_c, \mathbf{y}_c)$, using Kolmogorov's extension theorem (Oksendal, 2013), the collection of finite-dimensional distributions $\pi(\mathbf{y}_t; \mathbf{x}_c, \mathbf{y}_c, \mathbf{x}_t)$ for all $\mathbf{x}_{t,1}, \ldots, \mathbf{x}_{t,M} \in \mathbb{R}^M$, $M \in \mathbb{N}$, defines a stochastic process if these are consistent under (i) permutations of any entries of $(\mathbf{x}_t, \mathbf{y}_t)$ and (ii) marginalisations of any entries of $\mathbf{y}_t$. Prediction maps include, but are not limited to, Bayesian posteriors. One familiar example of such a map is the Bayesian Gaussian process (GP; Rasmussen, 2003) posterior

$$\pi(\mathbf{y}_t; \mathbf{x}_c, \mathbf{y}_c, \mathbf{x}_t) = \mathcal{N}(\mathbf{y}_t; \mathbf{m}, \mathbf{K}), \tag{2}$$

where $\mathbf{m} = m(\mathbf{x}_c, \mathbf{y}_c, \mathbf{x}_t)$ and $\mathbf{K} = k(\mathbf{x}_c, \mathbf{x}_t)$ are given by the usual GP posterior expressions. Another prediction map is the CNP Garnelo et al. (2018a):

$$\pi(\mathbf{y}_t; \mathbf{x}_c, \mathbf{y}_c, \mathbf{x}_t) = \prod_{m=1}^{M} p(y_{t,m}|\mathbf{r}_m), \tag{3}$$

where each $p(y_{t,m}|\mathbf{r}_m)$ is an independent Gaussian and $\mathbf{r}_m = r(\mathbf{x}_c, \mathbf{y}_c, x_{t,m})$ is parameterised by a DeepSet (Zaheer et al., 2017). CNPs are permutation and marginalisation consistent and thus correspond to valid stochastic processes. However, CNPs do not respect the product rule in general (Foong et al., 2020). Nevertheless, CNPs and their variants (Gordon et al., 2020) have been demonstrated to give competitive performance and robust predictions in a variety of tasks and are a promising class of meta-learning models.

**Gaussian Neural Processes:** A central problem with CNP predictive distributions is that they are mean-field: eq. (3) does not model correlations between $y_{t,m}$ and $y_{t,m'}$ for $m \neq m'$. However, many tasks require modelling dependencies in the output variable. To remedy this, we consider parameterising a correlated multivariate Gaussian

$$\pi(\mathbf{y}_t; \mathbf{x}_c, \mathbf{y}_c, \mathbf{x}_t) = \mathcal{N}(\mathbf{y}_t; \mathbf{m}, \mathbf{K}) \tag{4}$$

where, instead of the expressions for the Bayesian GP posterior, we use neural networks to parameterise the mean $\mathbf{m} = m(\mathbf{x}_c, \mathbf{y}_c, \mathbf{x}_t)$ and covariance $\mathbf{K} = K(\mathbf{x}_c, \mathbf{y}_c, \mathbf{x}_t)$. We refer to this class of models as Gaussian Neural Processes (GNPs). The first such model, the FullConvGNP, was introduced by Bruinsma et al. (2021) with promising results. Unfortunately, the FullConvGNP relies on $2D$-dimensional convolutions for parameterising $\mathbf{K}$, applying the sequence of computations

$$(\mathbf{x}_c, \mathbf{y}_c) \xrightarrow{\text{①}} (\tilde{\mathbf{x}}, \mathbf{h}) \xrightarrow{\text{②}} \mathbf{r} = \mathrm{PSD}(\mathrm{CNN}_{2D}(\mathbf{h})) \xrightarrow{\text{③}} \mathbf{K}_{ij} = \sum_{l=1}^{L} \psi(x_{t,i}, \tilde{x}_l) \, r_l \, \psi(\tilde{x}_l, x_{t,j}) \tag{5}$$

where ① maps $(\mathbf{x}_c, \mathbf{y}_c)$ to a $2D$-dimensional grid $\mathbf{h}$ at locations $\tilde{\mathbf{x}} = (\tilde{x}_1, \ldots, \tilde{x}_L)$, $\tilde{x}_l \in \mathbb{R}^{2D}$, using a SetConv layer (Gordon et al., 2020), ② maps $\mathbf{h}$ to $\mathbf{r}$ through a CNN with $2D$-dimensional convolutions, followed by a PSD map which ensures $\mathbf{r}$ is positive-definite, and ③ aggregates $\mathbf{r}$ using an RBF $\psi$. The CNN at ② requires expensive $2D$-dimensional convolutions, which are challenging to scale to higher dimensions (see appendix B). To overcome this difficulty, we propose parameterising $\mathbf{m}$ and $\mathbf{K}$ by

$$\mathbf{m}_i = f(x_{t,i}, \mathbf{r}), \quad \mathbf{K}_{ij} = k(g(x_{t,i}, \mathbf{r}), g(x_{t,j}, \mathbf{r})) \tag{6}$$

where $\mathbf{r} = r(\mathbf{x}_c, \mathbf{y}_c)$, $f$ and $g$ are neural networks with outputs in $\mathbb{R}$ and $\mathbb{R}^{D_g}$, and $k$ is an appropriately chosen positive-definite function. Note that, since $k$ models a posterior covariance, it cannot be stationary. The special case where $\mathbf{K}_{ij} = \sigma_i^2 \mathbf{I}_{ij}$ is diagonal corresponds to a mean-field CNP as presented in (Garnelo et al., 2018a). Equation (6) defines a class of GNPs which, unlike the FullConvGNP, do not require costly convolutions. GNPs can be readily trained via the log-likelihood

$$\theta^* = \arg\max_\theta \log \pi(\mathbf{y}_t; \mathbf{x}_c, \mathbf{y}_c, \mathbf{x}_t), \tag{7}$$

where $\theta$ collects all the parameters of the neural networks $f$, $g$, and $r$. In this work, we consider two methods to parameterise $\mathbf{K}$, which we discuss next.

**Linear covariance:** The first method we consider is the `linear` covariance

$$\mathbf{K}_{ij} = g(x_{t,i}, \mathbf{r})^\top g(x_{t,j}, \mathbf{r}) \tag{8}$$

which can be seen as a linear-in-the-parameters model with $D_g$ basis functions and a unit Gaussian distribution on their weights. This model meta-learns $D_g$ context-dependent basis functions, which approximate the true distribution of the target, given the context. By Mercer's theorem (Rasmussen, 2003), up to regularity conditions, every positive-definite function $k$ can be decomposed as

$$k(z, z') = \sum_{d=0}^{\infty} \phi_d(z)\phi_d(z') \tag{9}$$

where $(\phi_d)_{d=1}^{\infty}$ is a set of orthogonal basis functions. We therefore expect eq. (8) to be able to recover arbitrary (sufficiently regular) GP predictives as $D_g$ grows large. Further, the `linear` covariance has the attractive feature that sampling from it scales linearly with the number of target locations. A drawback is that the finite number of basis functions may limit its expressivity.

**Kvv covariance:** An alternative covariance which sidesteps this issue, is the `kvv` covariance

$$\mathbf{K}_{ij} = k(g(x_{t,i}, \mathbf{r}), g(x_{t,j}, \mathbf{r}))v(x_{t,i}, \mathbf{r})v(x_{t,j}, \mathbf{r}), \tag{10}$$

where $k$ is the Exponentiated Quadratic (EQ) covariance with unit lengthscale and $v$ is a scalar-output neural network. The $v$ modulate the magnitude of the covariance, which would otherwise not be able to shrink near the context. Unlike `linear`, `kvv` is not limited by a finite number of basis functions, but the cost of drawing samples from it scales cubically in the number of target points.

**Multi-output regression:** Extending this approach to the multi-output setting where $y_{t,m} \in \mathbb{R}^{D_y}$ with $D_y > 1$, can be achieved by learning functions $m_1, \ldots, m_{D_y}$ and $g_1, \ldots, g_{D_y}$ for each dimension of the output variable. We can represent covariances across different target points and different target vector entries, by passing those features through either the `linear` or the `kvv` covariance

$$\mathbf{K}_{ijab} = g_a(x_{t,i}, \mathbf{r})^\top g_b(x_{t,j}, \mathbf{r}), \tag{11}$$

$$\mathbf{K}_{ijab} = k(g_a(x_{t,i}, \mathbf{r}), g_b(x_{t,j}, \mathbf{r}))v_a(x_{t,i}, \mathbf{r})v_b(x_{t,j}, \mathbf{r}), \tag{12}$$

where $\mathbf{K}_{ijab}$ denotes the covariance between entry $a$ of $y_{t,i}$ and entry $b$ of $y_{t,j}$.

**Neural architectures:** This discussion leaves room for choosing $f$, $g$, and $r$, producing different models belonging to the *GNP family*, of which the FullConvGNP is also a member. For example, we may choose these to be DeepSets, attentive Deepsets or CNNs, giving rise to Gaussian Neural Processes (GNPs), Attentive GNPs (AGNPs) or Convolutional GNPs (ConvGNPs) respectively. Particularly, in the ConvGNP, the feature function $g$ takes the form

$$(\mathbf{x}_c, \mathbf{y}_c) \xrightarrow{①} (\tilde{\mathbf{x}}, \mathbf{h}) \xrightarrow{②} \mathbf{r} = \mathrm{CNN}_D(\mathbf{h}) \xrightarrow{③} g(x_{t,i}, \mathbf{r}) = \sum_{l=1}^{L} \psi(x_{t,i}, x_{r,l}) \, r_l, \tag{13}$$

where, crucially, $\mathbf{h}$ are values on a $D$-dimensional grid at $\tilde{\mathbf{x}} = (\tilde{x}_1, \ldots, \tilde{x}_L)$, $\tilde{x}_l \in \mathbb{R}^D$, and ② uses $D$-dimensional rather than a $2D$-dimensional CNN. This renders the ConvGNP much cheaper than the FullConvGNP in both compute and memory, while retaining translation equivariance (see appendix A.1 for proof), making the former a scalable alternative to the latter.

## 3 NON-GAUSSIAN PREDICTION MAPS

For many tasks joint-Gaussian models are sufficient. However, many tasks require non-Gaussian marginal distributions instead, for example because of non-negative or heavy-tailed variables.

**Gaussian Copula Neural Processes:** To address this issue, we draw inspiration from copula models (Elidan, 2013). Copulae use a dependent base distribution to model correlations, and adjust its marginals using invertible transformations to better approximate the data at hand. Wilson & Ghahramani (2010) and Jaimungal & Ng (2009) have extended copulae to the stochastic process setting, by using a GP as a base measure, and transforming its marginals appropriately. Adapting the approach of Jaimungal & Ng, we consider the following transformation to the output of a GNP

$$\mathbf{y}_t = \Phi_M^{-1}(\mathbf{u}_t, \boldsymbol{\psi}), \text{ where } \mathbf{u}_t = \Phi_G(\mathbf{v}_t) \text{ and } p(\mathbf{v}_t) = \pi_G(\mathbf{v}_t; \mathbf{x}_c, \mathbf{y}_c, \mathbf{x}_t). \tag{14}$$

where $\Phi_G$ is the CDF of the standard Gaussian, $\Phi_M^{-1}$ is the inverse CDF of a chosen distribution with parameters $\boldsymbol{\psi} = \psi(\mathbf{x}_c, \mathbf{y}_c, \mathbf{x}_t)$, and $\pi_G$ is a Gaussian prediction map. We refer to models of this form as Gaussian Copula Neural Processes (GCNPs). The log-likelihood of this model can be computed exactly using the change of variables formula (Kobyzev et al., 2020). Since the transformation in eq. (14) is dimension-wise, the resulting Jacobian is diagonal, and the log-likelihood takes the form

$$\log \pi(\mathbf{y}_t; \mathbf{x}_c, \mathbf{y}_c, \mathbf{x}_t) = \log \pi_G(\mathbf{v}_t; \mathbf{x}_c, \mathbf{y}_c, \mathbf{x}_t) + \sum_{m=1}^{M} \log |\Theta'(\mathbf{y}_{t,m})|, \tag{15}$$

where $\Theta(\cdot) = \Phi_G^{-1}(\Phi_M(\cdot, \boldsymbol{\psi}))$, $\mathbf{v}_t = \Theta(\mathbf{y}_t)$. Generally, $\Theta$ can be any arbitrary invertible map such as a Normalising Flow (NF). However, requiring marginalisation consistency places limitations on the form of $\Theta$. While point-wise NFs are consistent under marginalisations, it is unclear if non-marginal NFs can be used in a consistent model (appendix A.2). Last, while fully learnable marginal NFs (Durkan et al., 2019) can be used, the $\Theta$ presented here was sufficient for our experiments.

## 4 EXPERIMENTS

We apply the proposed models to synthetic and real data. Our experiments with synthetic data comprise four Gaussian tasks and a non-Gaussian task. In our experiments with real data, we evaluate our models on electroencephalogram data as well as climate data. We train our models and the Full-ConvGNP, whenever applicable, using the maximum-likelihood objective in eq. (7). We also train the ANP and ConvNP models as discussed in Foong et al. (2020). These are latent variable models which place a distribution $q$ over $\mathbf{r}$ and rely on $q$ for modelling output dependencies. Following Foong et al. we train the ANP and ConvNP via a biased Monte Carlo estimate of the objective

$$\theta^* = \arg\max_\theta \log\left[\mathbb{E}_{\mathbf{r}\sim q(\mathbf{r})}\left[p\left(\mathbf{y}_t; \mathbf{x}_c, \mathbf{y}_c, \mathbf{x}_t, \mathbf{r}\right)\right]\right]. \tag{16}$$

### 4.1 GAUSSIAN SYNTHETIC EXPERIMENTS

We apply the proposed models to synthetic datasets generated from GPs with four different covariance functions. In these experiments we have access to the ground truth predictive posterior, which we can use to assess the performance of our models. For each task, we generate multiple datasets using a fixed covariance function, and sub-sample these datasets into context and target sets, to which we fit the models (fig. 3). We consider tasks with one and two-dimensional inputs, the latter being a problem where the FullConvGNP cannot be feasibly applied to. Figures 4 and 5 compare the predictive log-likelihood of the models on these tasks, from which we observe the following trends.

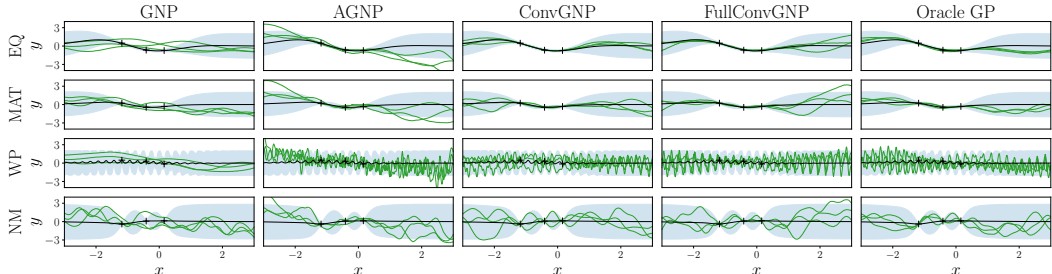

Figure 3: Samples drawn from the models' predictive posteriors (green) compared to the ground truth marginals (blue), using the `kvv` covariance.

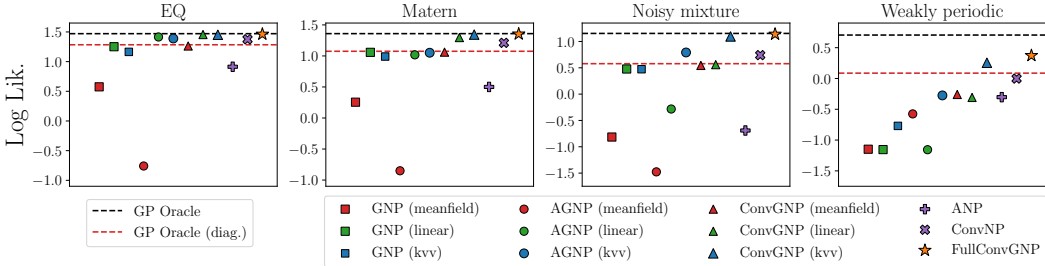

Figure 4: Predictive log-likelihoods across datasets for the 1D Gaussian tasks. The oracle GP performance is shown in dashed black. The dashed red line marks the performance of the diagonal GP oracle, where the off-diagonal covariance terms are 0. *Error bars too small to be seen in the plots.*

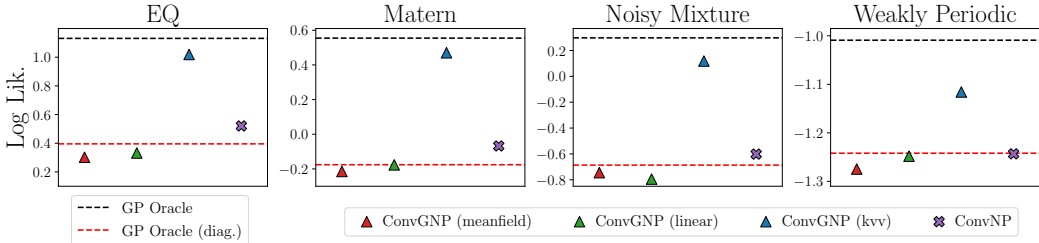

Figure 5: Predictive log-likelihood performance of the models, across datasets for the 2D Gaussian experiments, where the FullConvGNP is not applicable. *Error bars too small to be seen in the plots.*

**Dependencies improve performance:** We expected that modelling dependencies would allow the models to achieve better log-likelihoods. Indeed, for a fixed architecture, the correlated GNPs (■, ■, ●, ●, ▲, ▲) typically outperform their mean-field counterparts (■, ●, ▲). This suggests that our models can learn meaningful dependencies in practice, in some cases recovering oracle performance.

**Correlated ConvGNPs compete with the FullConvGNP:** The correlated ConvGNPs (▲, ▲) are often competitive with the FullConvGNP (★). The kvv ConvGNP (▲) is the only model, from those examined here, which competes with the FullConvGNP in all tasks. Unlike the latter, however, the former is scalable to higher input dimensions, and remains the best performing model in the two-dimensional tasks (fig. 5). For further details on the runtime and memory costs, see appendix B.

**Correlated GNPs outperform latent variable models:** Correlated GNPs typically outperform the latent-variable ANP (✚) and ConvNP (✖) models, which could be explained by the fact that the GNPs have a Gaussian predictive while ANP and ConvNP do not, and all tasks are Gaussian. Despite experimenting with different architectures, and even allowing for many more parameters in the ANP and ConvNP compared to the AGNP (●, ●) and ConvGNP (▲, ▲), we found it difficult to make the latent variable models competitive with correlated GNPs. We typically found the GNP family significantly easier to train than the latent variable models.

**`Kvv` outperformed `linear`:** We generally observed that the kvv models (■, ●, ▲) performed as well, and occasionally better than, their linear counterparts (■, ●, ▲). To test whether the linear models were limited by the number of basis functions $D_g$, we experimented with various settings $D_g \in \{16, 128, 512, 2048\}$. We did not observe a performance improvement for large $D_g$, suggesting that the models are not limited by this factor. This is surprising because, as $D_g \to \infty$ and assuming flexible enough $f$, $g$, and $r$, the linear models should, by Mercer's theorem, be able to recover any (sufficiently regular) GP posterior. We leave open the possibility that the linear models might be more difficult to optimise and thus struggle to compete with kvv.

**Predictive samples:** Figure 3 shows samples drawn from the predictives, from which we qualitatively observe that, like the FullConvGNP, the ConvGNP produces good quality function samples, which capture the behaviour of the underlying process. The ConvGNP is the only conditional model (other than the FullConvGNP) which produces high-quality posterior samples. The predictive log-likelihood (fig. 4) can be used as an objective measure for sample quality.

## 4.2 PREDATOR-PREY EXPERIMENTS

To assess performance on a non-Gaussian synthetic task, we generate data from a Lotka-Volterra predator-prey model (Arnold, 1992). This model describes the evolution of the populations of a predator and a prey species, which are related via a stochastic non-linear difference equation. These time series are non-Gaussian, since neither of the populations can fall below zero. To encode this prior knowledge, we use marginal transformations, enforcing non-negativity in the output variable.

We generate synthetic data following the method of Gordon et al. (2020) (see appendix D). We train and evaluate the ConvGNP, ConvNP and FullConvGNP models. We also train ConvGCNPs with an exponential CDF transformation $\Phi_M(u) = 1 - e^{-u/\psi}$, $\psi = \psi(\mathbf{x}_c, \mathbf{y}_c, \mathbf{x}_t)$. The marginals of these ConvGCNPs explicitly encode the prior knowledge that the populations are non-negative quantities. Figure 6 shows the model fits on a fixed predator time series. It also shows the predictive log-likelihood on test data and the log-likelihood on a downstream estimation task, explained below.

**Dependencies improve performance:** Similarly to the Gaussian tasks, we observe that modelling output dependencies (▲, ◆, ▲, ◆) improves the predictive log-likelihood over mean field models (▲, ◆). Further, the kvv covariance (▲, ◆) performs better than the linear covariance (▲, ◆).

**Marginal transformations:** We observe that exponential marginal transformations improve performance in the correlated models, as the ConvGCNP models (◆, ◆) typically exhibit better performance than their ConvGNP counterparts (▲, ▲). Further, the ConvGCNPs produce non-negative posterior samples, which are arguably more plausible than the samples produced the other models.

**Comparison with the ConvNP and FullConvGNP:** The ConvGNP with a kvv covariance (▲) is competitive with the FullConvGNP (★), while adding marginal transformations can further improve the model's performance (◆). Both models outperform the ConvNP (✖) by a considerable margin.

**Correlated GNPs and downstream estimation:** We assessed the performance of the models on a downstream estimation task, illustrated in fig. 7. Given a context set, we use the trained models

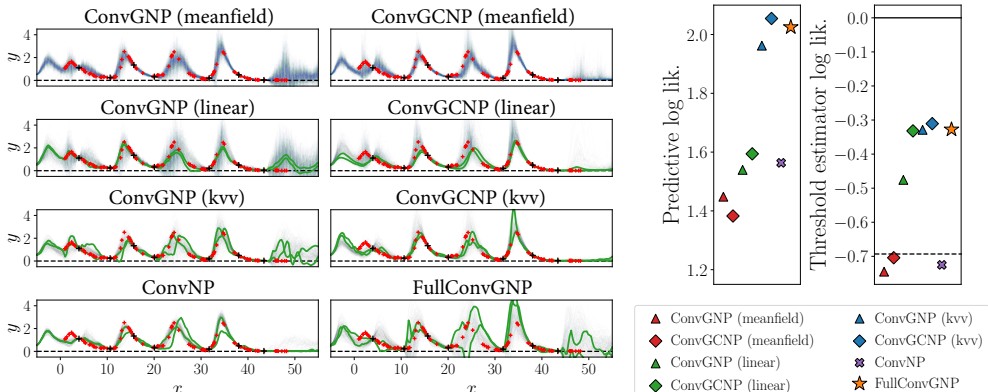

Figure 6: The predator modelling task. Model fits (left) where black and red crosses show the context and target sets of a dataset, the blue regions show the marginals, and the green lines are samples from the predictive. The dashed line marks $y = 0$. *Error bars for per-datapoint predictive log-likelihoods and threshold estimation task log-likelihoods too small to be seen in the plots.*

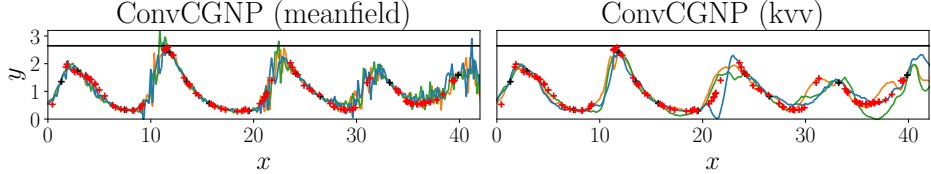

Figure 7: Illustration of the failure mode of mean-field models in the threshold estimation task. The context and target are shown in black and red crosses, and the black line shows the threshold for this context set. In each plot, three samples from the predictive are shown in orange, green and blue.

to estimate the probability that the population will exceed the maximum population observed in the context, by a factor of at least $1.1$. Mean-field models make independent predictions, and thus the proportion of function samples which exceed the threshold is unreasonably large. By contrast, a correlated model avoids this failure mode as their function samples are coherent. Figure 6 shows that correlated models (▲, ◆, ▲, ◆, ⬟) are substantially more accurate than mean-field models (▲, ◆) in this threshold estimation task, with the latter being marginally worse than a random prediction.

## 4.3 ELECTROENCEPHALOGRAM EXPERIMENTS

We applied the proposed and competing models on a real electroencephalogram (EEG) dataset. This dataset comprises of 7632 multivariate time series, collected by placing electrodes on different subjects' scalps (Zhang et al., 1995). Each time series exhibits correlations across channels, as the levels of activity at different regions of the subjects' brain are correlated. It is plausible that there also exist patterns which are shared between different time series from a single subject, as well as across subjects, making this an ideal task for a multi-output meta-learning model such as the ConvGNP. We showcase the ability of the ConvGNP to perform correlated multi-output predictions and handle missing data. We train the models on time series with 7 channels, where we occlude randomly chosen windows of 3 of the channels, using the occluded values as targets. Table 1 shows the models' performance on held-out test data, and fig. 8 shows an example fit of a ConvGNP. To demonstrate the benefits of meta-learning, the table includes a multi-output GP baseline (MOGP, Bruinsma et al., 2020), which is trained from scratch for every task, without a meta-learning component (see appendix E). Observe that this baseline is significantly outperformed by all meta-models.

**Correlated models improve predictive log-likelihood:** We observe that the correlated ConvGNPs show a considerable improvement in terms of log-likelihood, compared to the mean-field ConvGNP as well as the ConvNP. This improvement in log-likelihood suggests that the correlated ConvGNPs learn to represent meaningful correlations in the target outputs.

**ConvGNPs can model multivariate and partially observed data:** Figure 8 shows that the `kvv` ConvGNP is able to infer unobserved values from the EEG data of a held-out test example. The model produces both calibrated marginals as well as plausible function samples, and represents not only temporal, but also cross-channel correlations.

| | MEAN-FIELD | CONVGNP linear | kvv | CONVNP | MOGP |
|---|---|---|---|---|---|
| LOG LIK. | $-5.27 \pm 0.01$ | $\mathbf{-1.39 \pm 0.01}$ | $\mathbf{-1.24 \pm 0.00}$ | $-3.96 \pm 0.01$ | $-12.7 \pm 0.42$ |

Table 1: Per-datapoint log-likelihood on the held-out EEG test set.

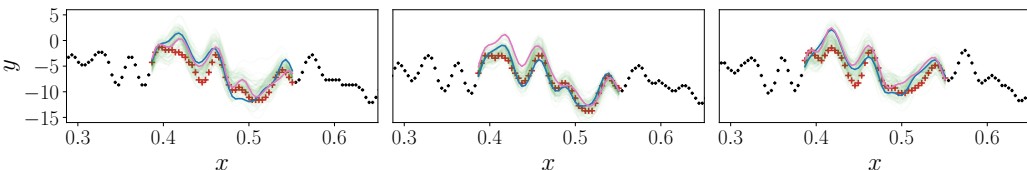

Figure 8: Fit of the ConvGNP (kvv) on the EEG data. Each pane shows one of the three channels with unobserved targets (red crosses). All other data (black crosses), including the remaining four channels are observed. Marginals are shown in green, and two samples are shown in blue and pink.

### 4.4 TEMPERATURE DOWNSCALING FOR ENVIRONMENTAL MODELLING

Lastly, we apply the models on a environmental modelling task. In climate modelling, future projections are obtained by simulating the atmospheric equations of motion on a spatio-temporal grid. Unfortunately, computational constraints typically limit spatial resolution to around 100-200km (Eyring et al., 2016), which is insufficient to resolve extreme events and produce local projections (Allen et al., 2016; Maraun et al., 2017). To address this issue, so-called *statistical downscaling* is routinely applied. A mapping from low-resolution simulations to historical data is learnt, and is then applied to future simulations to produce high-resolution predictions (Maraun & Widmann, 2018).

While numerous data-driven approaches to statistical downscaling exist, they are often limited to making predictions on a fixed set of points at which historical observations are available (Sachindra et al., 2018; Vandal et al., 2017; 2018; 2019; Bhardwaj et al., 2018; Misra et al., 2018; White et al., 2019; Pan et al., 2019; Baño-Medina et al., 2020; Liu et al., 2020). Recently, Vaughan et al. (2021) have applied ConvCNPs to temperature downscaling, enabling predictions at arbitrary locations. Though the ConvCNP outperforms an ensemble of existing methods, it is unable to generate coherent samples, limiting its practical applicability.

**Experimental setup:** We modify the model of Vaughan et al. (2021), which maps a low-resolution grid of reanalysis data together with local orographic information, to a set of features used to parameterise a Gaussian predictive. We use reanalysis and historical station data throughout Europe, to set up three experiments, where we train on data from the years 1979-2002, and make predictions on a held out test set from the years 2003-2008, emulating realistic prediction tasks using future climate simulations. The first experiment uses a training set of 86 specific stations in Europe, following a standardised experimental protocol known as the VALUE framework (Maraun et al., 2015), for which extensive baselines are available. Note that we train and test the models on the same stations, but at different time periods. The second is a larger experiment consisting of a training set of 3043 stations across Europe and the 86 VALUE stations as a held out test set. The third experiment uses 713 training and 250 held out test stations, all located in Germany, which has densest station coverage in Europe. This experiment highlights the benefits of correlated models, since the stations are near one another and are thus highly correlated. For further details on our setup, see appendix F.

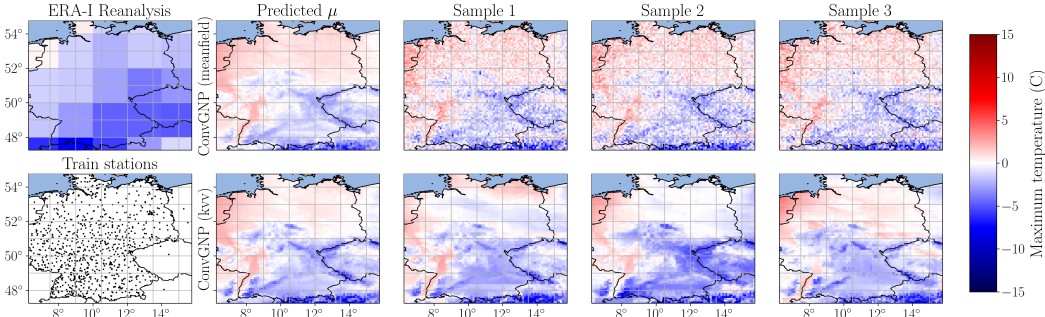

Figure 9: Illustration of sampled temperature fields. After training on low-res. simulations (top left) and observed data (bottom left), the models are conditioned on future low-res. simulations, to make predictions. The remaining columns show the predicted mean and three samples from the predictive.

**Correlations improve performance:** Across all three experiments, modelling correlations improves the predictive log-likelihood (table 2). Since small differences are observed in the MAE, we conclude that the correlated models have learnt to model meaningful statistical output dependencies. The log-likelihood improvement is greatest for the Germany experiment, where the stations are near one another and thus more strongly correlated. In the VALUE experiment, the correlated ConvGNPs improve on the ConvNP, as well as on the mean-field ConvGNP, which has itself been shown (Vaughan et al., 2021) to outperform an ensemble of widely used statistical downscaling methods.

**Sampling coherent temperature fields:** Correlated ConvGNP models can be used to sample coherent temperature fields, something that mean-field models cannot perform. Figure 9 shows that the sampled fields are not only coherent, but also exhibit non-trivial correlation structure.

|  | EUROPE (VALUE) | EUROPE (ALL) | GERMANY |
|---|---|---|---|
| CONVGNP (MEAN-FIELD) | $-148.12$ / $1.06$ | $-176.93$ / $1.41$ | $-263.90$ / $0.94$ |
| CONVGNP (LINEAR) | $-131.58$ / $1.03$ | $\mathbf{-153.23}$ / $1.33$ | $-224.74$ / $\mathbf{0.92}$ |
| CONVGNP (KVV) | $\mathbf{-131.47}$ / $1.02$ | $-157.25$ / $\mathbf{1.30}$ | $\mathbf{-209.06}$ / $1.00$ |
| CONVNP | $-144.14$ / $1.08$ | $-163.11$ / $1.38$ | $-252.94$ / $1.13$ |
| VALUE BASELINE (MAE) | $1.32$ | | |

Table 2: Predictive log-likelihood and MAE for the three temperature prediction experiments.

## 5 RELATED WORK

**Conditional and Latent Neural Processes:** Our work proposes novel members of the CNP family (Garnelo et al., 2018a; Gordon et al., 2020), which capture output dependencies, and also lend themselves to multi-output and non-Gaussian predictions. Similar approaches which model output correlations include the FullConvGNP model of Bruinsma et al. (2021) which, however, cannot be feasibly scaled beyond one-dimensional input data, due to its use of $2D$-dimensional convolutions. The latent Neural Process family (Garnelo et al., 2018b; Kim et al., 2019; Foong et al., 2020) is another model class that can capture output dependencies. Unfortunately, these models involve latent variables which make the log-likelihood analytically intractable and complicate training (Le et al., 2018). Petersen et al. (2021) recently introduced a model similar to the ConvNP, which uses a GP to represent the latent variable of a ConvNP-like model, however this too suffers from the same training challenges, and has a computational cost that scales cubically with the context size.

**Deep Kernels:** The GNP covariances we presented bear similarities to Deep Kernels (DKs; Patacchiola et al., 2019; Calandra et al., 2016; Wilson et al., 2015). DKs also use neural networks in the context of GPs but, unlike GNPs which define a GP *predictive*, DKs define a GP *prior*, which is conditioned on data using Bayes' rule. Thus, the computational cost of DKs at test-time is cubic in the context points, whereas that of GNPs is linear, enabling the latter to scale to larger datasets.

**Copula Processes:** Our ConvCGNP models can be regarded as meta-learning versions of the Copula Processes of Wilson & Ghahramani (2010) and Jaimungal & Ng (2009). These transform the marginals of a GP, defining a non-Gaussian prior process, and perform inference over it, which requires compute which scales cubically with the size of the context. By contrast, ConvGCNPs directly parameterise a predictive, making their test-time complexity linear in the context.

## 6 CONCLUSION

In this work, we present a tractable method for modelling statistical dependencies in the output of a CNP. We propose parameterising the covariance of a predictive Gaussian by passing context-dependent feature vectors through positive definite covariance functions. The resulting GNP models, account for output correlations, but unlike existing methods, they can be applied to data with higher-dimensional inputs while maintaining an analytically tractable log-likelihood, which makes them especially easy to train. GNPs can be extended to multi-output regression, and also combined with invertible marginal transformations to model non-Gaussian data. We demonstrate that modelling correlations improves predictive performance over mean-field models on Gaussian and non-Gaussian synthetic data, including a downstream estimation task that mean-field models cannot solve. Our models also show improved performance over their mean-field and latent counterparts on real EEG and climate tasks. In statistical temperature downscaling, our models outperform a standard ensemble of widely used methods in statistical downscaling, while providing spatially coherent temperature samples. This renders our models suitable for application to climate impact studies.

## 7 ETHICS STATEMENT

We believe that modelling statistical output dependencies with GNPs will enable members of the Neural Process family to be used as part of larger estimation pipelines, for example, but not limited to, climate modelling pipelines, to improve human decision making. However, since the effectiveness of meta-learning models depends on the amount and quality of training data available (Bronskill, 2020), we acknowledge that relying on over- or under-estimated uncertainties produced by GNPs, may cause adverse outcomes. Training procedures which do not safeguard against these pitfalls may result in mis-estimation of statistical dependencies which may have serious practical effects, such as bias and discrimination. This concern however is not unique to the GNP family, or to meta-learning, but in virtually every uncertainty-aware probabilistic model. We believe that acknowledging and investigating these issues is useful for producing reliable probabilistic models, to improve our decision making processes.

## 8 REPRODUCIBILITY

All our experiments are carried out using either synthetic or publicly available datasets. We provide precise details for how to generate the data used for the synthetic experiments in appendices C and D. The EEG data is available via the UCI dataset website (link[1]). The climate modelling data is also available to the public, through the Copernicus European Climate Data Store (link[2]).

We also provide the details of our models, such as architectural choices, and the training schemes used in each experiment in appendices C to F. We publish the complete repository containing all our code, including the models, training scripts, pretrained models and Jupyter notebooks which will produce all plots in this paper (link[3]).

## 9 ACKNOWLEDGEMENTS

We thank David Duvenaud for helpful comments on a draft. Wessel P. Bruinsma was supported by the Engineering and Physical Research Council (studentship number 10436152). Anna Vaughan acknowledges the UKRI Centre for Doctoral Training in the Application of Artificial Intelligence to the study of Environmental Risks (AI4ER), led by the University of Cambridge and British Antarctic Survey, and studentship funding from Google DeepMind. Richard E. Turner is supported by Google, Amazon, ARM, Improbable and EPSRC grant EP/T005386/1.

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

# A    ADDITIONAL THEORETICAL CONSIDERATIONS

In this appendix we provide some additional discussion on theoretical aspects of our models. In appendix A.1 we provide a proof for the translation equivariance of the ConvGNP, which follows from the fact that the ConvCNP is translation equivariant. In appendix A.2 we provide some additional discussion on composing arbitrary invertible maps such as Normalising Flows (NFs) with neural processes.

## A.1    TRANSLATION EQUIVARIANCE OF THE CONVGNP

In this section of the appendix we provide a proof for the translation equivariance of the ConvGNP model. The mean function of a ConvGNP has the same form as a ConvCNP model (Gordon et al., 2020) and is thus translation equivariant. It therefore remains to show that the covariance function of the ConvGNP is also translation equivariant.

As explained in the main text, the covariance of the ConvGNP is computed as follows. First, a feature function $g$ is applied to the context set and target inputs. This feature function consists of the following sequence of computations

$$(\mathbf{x}_c, \mathbf{y}_c) \xrightarrow{\text{①}} (\tilde{\mathbf{x}}, \mathbf{h}) \xrightarrow{\text{②}} \mathbf{r} = \text{CNN}_D(\mathbf{h}) \xrightarrow{\text{③}} g(x_{t,i}, \mathbf{r}) = \sum_{l=1}^{L} \psi(x_{t,i}, x_{r,l}) \, r_l, \qquad (17)$$

where step ① maps $(\mathbf{x}_c, \mathbf{y}_c)$ to a $D$-dimensional grid of values $\mathbf{h}$ with corresponding locations at $\tilde{\mathbf{x}} = (\tilde{x}_1, \ldots, \tilde{x}_L)$, $\tilde{x}_l \in \mathbb{R}^D$, using a SetConv layer (Gordon et al., 2020), step ② maps $\mathbf{h}$ to $\mathbf{r}$ through a CNN with $D$-dimensional convolutions, and ③ aggregates $\mathbf{r}$ using an RBF $\psi$. The grid locations $\tilde{\mathbf{x}}$ are set using the context and target inputs (see Gordon et al., 2020), according to

$$\tilde{\mathbf{x}} = \text{grid}(x_{min}, x_{max}), \text{ where } x_{min} = \min\{\mathbf{x}_c, \mathbf{x}_t\}, x_{max} = \max\{\mathbf{x}_c, \mathbf{x}_t\}. \qquad (18)$$

Lastly, the features outputted by $g$ are passed through a positive definite function $k$ to produce the entries of the covariance

$$\mathbf{K}_{ij} = k(g(x_{t,i}, \mathbf{r}), g(x_{t,j}, \mathbf{r})). \qquad (19)$$

To show the translation equivariance of the covariance function, it suffices to show that if a translation is applied to both the context and target inputs, then the resulting covariance matrix remains unchanged. In particular, consider applying a translation $u$ to the context and target inputs

$$\mathbf{x}'_c = (x'_{c,1}, \ldots, x'_{c,N}) = (x_{c,1} + u, \ldots, x_{c,N} + u) \qquad (20)$$
$$\mathbf{x}'_t = (x'_{t,1}, \ldots, x'_{t,M}) = (x_{t,1} + u, \ldots, x_{t,M} + u) \qquad (21)$$

and applying $g$ to the translated inputs, according to

$$(\mathbf{x}'_c, \mathbf{y}_c) \xrightarrow{\text{①}} (\tilde{\mathbf{x}}', \mathbf{h}') \xrightarrow{\text{②}} \mathbf{r}' = \text{CNN}_D(\mathbf{h}') \xrightarrow{\text{③}} g(x'_{t,i}, \mathbf{r}') = \sum_{l=1}^{L} \psi(x'_{t,i}, x'_{r,l}) \, r'_l. \qquad (22)$$

where the grid locations are now $\tilde{\mathbf{x}}' = (\tilde{x}'_1, \ldots, \tilde{x}'_L)$, $\tilde{x}_l \in \mathbb{R}^D$, where by substituting $\mathbf{x}'_c, \mathbf{x}'_t$ into eq. (18) we have

$$\tilde{\mathbf{x}}' = \tilde{\mathbf{x}} + u. \qquad (23)$$

Now, noting that $\mathbf{h}' = \mathbf{h}$, by the translation equivariance of the SetConv layer (Gordon et al., 2020) and $\mathbf{r}' = \mathbf{r}$, by the equivariance of the CNN, and substituting these together with eqs. (20), (21) and (23) into eq. (22) we obtain

$$g(x'_{t,i}, \mathbf{r}') = \sum_{l=1}^{L} \psi(x'_{t,i}, x'_{r,l}) \, r'_l = \sum_{l=1}^{L} \psi(x_{t,i} + u, x_{r,l} + u) \, r_l = \sum_{l=1}^{L} \psi(x_{t,i}, x_{r,l}) \, r_l = g(x_{t,i}, \mathbf{r})$$

where we have used the stationarity of the RBF $\psi$. This shows that $g$ is invariant to translations of the context and target inputs. The covariance function of the ConvGNP is therefore equivariant, as required.

## A.2 NORMALISING FLOWS AND GENERAL INVERTIBLE MAPS

**Composing Normalising Flows with Neural Processes:** In section 3 of the main text we explained that general arbitrary maps could be composed with neural processes in an attempt to model joint dependencies and non-Gaussian marginals. However, requiring a model that is consistent under marginalisation places theoretical limitations on the form of $\Theta$ which we can use.

**Theoretical limitations:** Suppose we have a Neural Process with prediction map $\pi(\,\cdot\,; x_c, y_c, x_t)$, which we wish to compose with an invertible map $\Theta$. Thus, to draw a sample from the predictive distribution, we first draw

$$(u_{t,1}, \ldots, u_{t,M}) \sim \pi(u_t; x_c, y_c, x_t), \tag{24}$$

and then apply the marginal transformation

$$(y_{t,1}, \ldots, y_{t,M}) = \Theta(u_{t,1}, \ldots, u_{t,M}). \tag{25}$$

The role of $\Theta$ could be either modelling the marginals of the data, representing joint dependencies in the output variable, or both. Now, $\Theta$ must be able to handle variable-length tuples as arguments, to handle arbitrary target points. Thus, we should be able to query $\Theta$ with any number $M$ of inputs. In addition, the model should be consistent under marginalisations: making a joint prediction for target variables $y_{t,1}$ and $y_{t,2}$, and then marginalising over $y_{t,2}$, should be equivalent to making a prediction over $y_{t,1}$ directly. Put concretely, marginalisation consistency requires that the following equality in distribution holds

$$\Theta(u_{t,1}, u_{t,2})_i \overset{d}{=} \Theta(u_{t,i}). \tag{26}$$

For a careful choice of $\Theta$ and distribution of $(u_{t,1}, u_{t,2})$ this equality can be true, but in general will not be true. If we want to construct an invertible $\Theta$ such that it gives a consistent model for *all* underlying Neural Processes, then that requires the the stronger condition that

$$\Theta(u_{t,1}, u_{t,2})_i = \Theta(u_{t,i}) \quad \text{almost surely.} \tag{27}$$

From this condition, we see that $\Theta$ can only be a marginal transformation because

$$\Theta(u_{t,1}, u_{t,2}) = (\Theta(u_{t,1}), \Theta(u_{t,2})), \tag{28}$$

and similarly for other $M$. Though for a given Neural Process there may exist an appropriate $\Theta$ which satisfies marginalisation consistency, constraining $\Theta$ to achieve this is a challenging research problem, which we found to be beyond the scope of this paper.

**Marginal maps cannot model joint dependencies:** We now note that if $\pi$ is mean-field and $\Theta$ is a marginal transformation, then it follows that the $y_t$ variables are independent. Therefore, it is not possible to compose a mean-field CNP with an NF to model joint dependencies in this way. Instead, we must rely on $\pi$ for modelling dependencies, and $\Theta$ for learning the marginals.

## B COMPUTATION TIME AND MEMORY COMPARISON

**Forward pass for one-dimensional models:** In this section we provide quantitative measurements for the computational and memory costs of the models. Table 3 shows the runtime and memory footprint of the GNP, AGNP, ConvGNP and FullConvGNP models applied to data with one-dimensional inputs, $D = 1$. In particular, we measure the runtime and memory required to perform a single forward pass through the neural architecture of each model that was used for the one-dimensional Gaussian tasks, on an NVIDIA GeForce RTX 2080 Ti GPU.

**The ConvGNP is cheaper than the FullConvGNP:** In table 3 we see that the FullConvGNP model requires a factor of two more runtime and a factor of five larger memory footprint for this example. It should be noted that the performance difference between the ConvGNP and FullConvGNP is less pronounced in the one-dimensional setting, compared to the higher-dimensional settings for two reasons. First, both the runtime and memory footprint of the FullConvGNP increase much quicker than those of the ConvGNP, due to the fact that the FullConvGNP uses $2D$-dimensional convolutions. Second, the parallelisation of operations in the GPU may wash out some of the differences in runtime in favour of the FullConvGNP. More specifically, in cases where some of the GPU workers are idle, i.e. the GPU is not at its parallelisation limit, additional computations can be carried out at a small overhead by using these idle workers. Some of the additional computations required by the

|  | GNP | AGNP | ConvGNP | FullConvGNP |
|---|---|---|---|---|
| Runtime ($\times 10^{-3}$ sec) | $0.58 \pm 0.00$ | $1.62 \pm 0.00$ | $1.74 \pm 0.00$ | $4.59 \pm 0.01$ |
| Memory (MB) | 0.476 | 2.971 | 0.231 | 1.23 |

Table 3: Computational and memory costs for one-dimensional models, during a forward using a batch size of one, that is, a single task was passed to each model.

FullConvGNP come at a reduced runtime cost to what we would expect if the parallelisation limit of the GPU was reached. Since we expect this limit to be reached for larger $D$, we also expect the performance difference to be more pronounced for higher dimensions, for this reason.

**Forward pass through a CNN:** To further highlight the how the runtime and memory costs scale with the convolution dimension $D_c$, we measured the runtime and memory footprint of a CNN with $D_c = 1, 2$ and $3$. In particular we used a depth-wise separable CNN (Chollet, 2017) similar to that used in Bruinsma et al. (2021), consisting of twelve hidden convolutional layers in $D_c = 1, 2$ or $3$ dimensions, each with a kernel size of 5, measuring the runtime and memory cost of a forward pass through the network. Thus for each dimension, the CNN is applied to a tensor with $D_c$ dimensions, each with a size of $N = 128$, plus an additional channel dimension which we set to 2. Figure 10 shows that the runtime and memory required by the CNN increase exponentially with $D_c$. Extrapolating to $D_c = 4$, we observe that the runtime and memory costs become extremely large, making the FullConvGNP difficult to apply, even to data in $D = 2$ dimensions, since the FullConvGNP requires $D_c = 2D$ dimensional convolutions. For $D = 3$ the FullConvGNP would require $D_c = 6$ dimensional convolutions, which is would be significantly above the runtime and memory we can afford with existing GPUs.

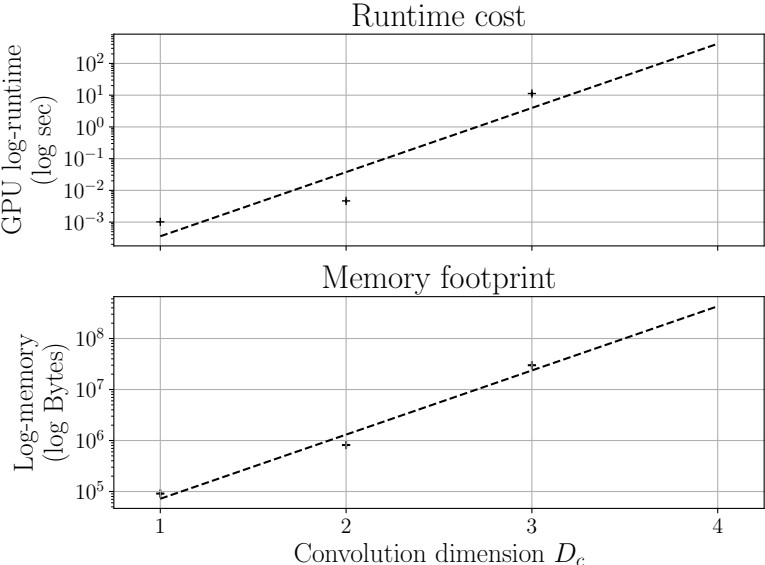

Figure 10: Scaling of the runtime cost and memory footprint of a CNN as a function of the convolution dimension. See text for discussion. *Error bars have been included for the runtime, but are too small to be seen in this plot.*

## C  Gaussian synthetic experiments

**Data generation process:** Each synthetic task consists of a collection of datasets sampled from the same generative process. To generate each of these datasets, we first determine the number of context and target points. We use a random number between 3 and 50 of context points and a fixed number of 100 target points. For each dataset we sample the inputs of both the context and target points, that

is $\mathbf{x}_c, \mathbf{x}_t$ uniformly at random in the region $[-2, 2]$ for the 1D tasks and in $[-2, 2] \times [-2, 2]$ for the 2D tasks. We then sample the corresponding outputs $\mathbf{y}_c, \mathbf{y}_t$ as follows.

**Exponentiated Quadratic (EQ):** We sample $\mathbf{y}_c, \mathbf{y}_t$ from a GP with an EQ covariance

$$k_{\text{EQ}}(x, x') = \sigma_v^2 \exp\left(-\frac{1}{2\ell^2}(x - x')^2\right),$$

with parameters $(\sigma_v^2, \ell) = (1.00, 1.00)$.

**Matern 5/2:** We sample $\mathbf{y}_c, \mathbf{y}_t$ from a GP with a covariance

$$k_{\text{M}}(x, x') = \sigma_v^2 \left(1 + \frac{r}{\ell} + \frac{r^2}{3\ell^2}\right) \exp\left(-\frac{r}{\ell}\right), \tag{29}$$

where $r = |x - x'|$, with parameters $(\sigma_v^2, \ell) = (1.00, 1.00)$.

**Noisy mixture:** We sample $\mathbf{y}_c, \mathbf{y}_t$ from a GP which is a sum of two EQ kernels

$$k_{\text{NM}}(x, x') = k_{EQ,1}(x, x') + k_{EQ,2}(x, x'),$$

with the following parameters $(\sigma_{v,1}^2, \ell_1) = (1.00, 1.00)$ and $(\sigma_{v,2}^2, \ell_2) = (1.00, 0.25)$.

**Weakly periodic:** We sample $\mathbf{y}_c, \mathbf{y}_t$ from a GP which is the product of an EQ and a periodic covariance

$$k_{\text{WP}}(x, x') = k_{EQ}(x, x') \exp\left(-\frac{2\sin^2(\pi|x - x'|/p)}{\ell_p^2}\right),$$

with EQ parameters $(\sigma_{v,EQ}^2, \ell_p) = (1.00, 1.00)$ and periodic parameters $(p, \ell_{EQ}) = (0.25, 1.00)$.

Lastly, for all tasks we add iid Gaussian noise with zero mean and variance $\sigma_n^2 = 0.05^2$. This noise level was not given to the models, which in every case learned a noise level from the data. We generate training data for 100 epochs, each consisting of 1024 iterations, each of which contains 8 different tasks. For testing, we use a single epoch of 1024 iterations, each of 8 different tasks.

**Neural architectures for 1D tasks (GNP, AGNP, ANP):** For the GNP model we use a fully connected neural network consisting of an encoder with six hidden layers of 128 units each, a mean aggregation layer and a decoder with a single hidden layer, also with 128 units. For the AGNP model, we use the same architecture as for the GNP model, except the aggregation layer consists of a dot product self-attention layer (Vaswani et al., 2017). For the ANP model, we follow the same architecture which was used in Kim et al. (2019), which uses a *deterministic* and a *stochastic* path in the encoder. The deterministic path consists of the same encoder used in the GNP and AGNP models, while the stochastic path consists of a fully connected network with two hidden layers of 128 units each. The outputs of the deterministic and the stochastic paths are concatenated and passed through the same decoder architecture as for the GNP and the AGNP.

**Neural architectures for 1D Gaussian tasks (ConvGNP, ConvNP):** We use the same architecture for all the ConvGNP models. This consists of a SetConv layer (Gordon et al., 2020), followed by a pointwise linear transformation with a nonlinearity, a CNN with a UNet architecture (Ronneberger et al., 2015), another linear transformation, and lastly a SetConv layer for producing the features required by the mean-field, `linear` or `kvv` covariance. The first SetConv maps the context set to a discretised grid with 64 points per unit, producing two channels at each point on the grid, referred to as the data and the density channels in Gordon et al. (2020). The pointwise linear transformation maps the two features outputted by the first SetConv to eight features which are fed to the UNet network. The UNet itself consists of six regular convolution layers with a kernel size of 5 and a stride of 2, and channel sizes

$$(c_{\text{in}}, c_{\text{out}}) = (8, 8), (8, 16), (16, 16), (16, 32), (32, 64),$$

followed by six layers of transpose convolutions, again with a kernel size of 5, a stride of 2 and channel sizes

$$(c_{\text{in}}, c_{\text{out}}) = (64, 32), (64, 32), (64, 16), (32, 16), (32, 8), (16, 8).$$

Note the numbers of channels in the transpose convolutions are of the above dimensions because of the additional connections of the UNet architecture. For the ConvNP model, we use a similar

architecture as for the ConvGNP, except the model contains two stacked UNet networks. The first SetConv of the ConvNP uses the same discretisation of 64 points per unit, as the ConvGNP. Then, the first UNet of the ConvNP is the same as the UNet of the ConvGNP, except it has twice the number of output channels. Half of these are used as the mean and the other half as the log-variance of 64 independent Gaussian variables, for each position in the convolution grid. This is followed by another UNet with the same architecture as the ConvGNP UNet, followed by a linear transformation and a SetConv layer. The SetConv maps the outputs of the second UNet and a target input to a mean and a log-variance.

**Neural architecture for 1D tasks (FullConvGNP):** For the FullConvGNP we follow Bruinsma et al. (2021), who use a one-dimensional ConvCNP-like for the predictive mean and another two-dimensional convolutional architecture for the predictive covariance. For the precise algorithm of the FullConvGNP, we refer the reader to Appendix E.2 of Bruinsma et al.. Here we give the specific details of the CNNs used in our implementation. Unlike Bruinsma et al., we use UNet-style architectures for both these CNNs, as opposed to depthwise-separable CNNs, because we find the former to train much quicker both in terms of wall-clock time as well as number of epochs. For the mean parametrisation, we use the same architecture as for the ConvGNP, except this outputs a single feature representing the predictive mean. For the covariance architecture we use a discretisation of 30 points per unit use a UNet consisting of six layer of two-dimensional convolutions, with a kernel size of $5 \times 5$, a stride of 2 and the same numbers of channels that are used in the mean parameterisation, namely

$$(c_{\text{in}}, c_{\text{out}}) = (8, 8), (8, 16), (16, 16), (16, 32), (32, 64),$$

followed by six layers of transpose convolutions, also with a kernel size of $5 \times 5$ and stride of 2, and channel sizes

$$(c_{\text{in}}, c_{\text{out}}) = (64, 32), (64, 32), (64, 16), (32, 16), (32, 8), (16, 8).$$

**Neural architectures for 2D tasks (ConvGNP, ConvNP):** We use the same UNet architectures for the 2D tasks as those for the 1D tasks, except the convolutions are now two-dimensional and the discretisation resolution of the SetConv layer is set to 32 points per unit.

**General notes:** We use the same number of features in the output layer of the GNP, AGNP and ConvGNP models. The mean-field covariance uses two features, one for the mean and one for the marginal variance of the predictive, while the `linear` and `kvv` models both use the same number of $D_g = 512$ features for computing the predictive covariance. We use a learnable homoscedastic noise variable for all models, and ReLU activation functions for all hidden layers. We optimise all models with Adam (Kingma & Ba, 2014), using a learning rate of $5 \times 10^{-4}$. For the ConvNP we use 10 latent samples to evaluate the loss during training, and 512 samples during testing. We do not use any weight regularisation.

## D   PREDATOR-PREY SYNTHETIC EXPERIMENTS

**Data generation process:** We broadly follow the method specified in Appendix C.4 of Gordon et al. (2020) for generating data from the Lotka-Volterra model, which uses the algorithm specified in Gillespie (1977). For each time series, we first sample the predator birth and death parameters $\theta_1, \theta_2$ and the $\theta_3, \theta_4$ from the distributions

$$\theta_1 \sim 1.00 \times 10^{-2} \times \text{Uniform}(1 - \epsilon, 1 + \epsilon), \tag{30}$$

$$\theta_2 \sim 5.00 \times 10^{-1} \times \text{Uniform}(1 - \epsilon, 1 + \epsilon), \tag{31}$$

$$\theta_3 \sim 5.00 \times 10^{-1} \times \text{Uniform}(1 - \epsilon, 1 + \epsilon), \tag{32}$$

$$\theta_4 \sim 1.00 \times 10^{-2} \times \text{Uniform}(1 - \epsilon, 1 + \epsilon), \tag{33}$$

where $\epsilon = 0.1$. We choose these parameters following Gordon et al. (2020), because they result in plausible oscillatory as well as transient behaviours of the predator and prey populations. We then initialise the predator and prey populations at $X = 50$ and $Y = 100$ respectively at time $t = 0$, and perform a sequence of discrete steps. At each step one of the following events occur:

1. We sample $\Delta t \sim \text{Exponential}(R^{-1})$, where $R = \theta_1 XY + \theta_2 X + \theta_3 Y + \theta_4 XY$.

2. We sample one of the following events:

    (a) A predator is born with probability $\theta_1 XY/R$, increasing $X$ by 1.

    (b) A predator dies with probability $\theta_2 X/R$, decreasing $X$ by 1.

    (c) A prey is born with probability $\theta_3 Y/R$, increasing $Y$ by 1.

    (d) A prey dies with probability $\theta_4 XY/R$, decreasing $Y$ by 1.

3. Increment $t \leftarrow t + \Delta t$ and repeat until $t = 100$ is reached, or when $10^4$ events occur.

Lastly, we linearly interpolate the predator and prey time series and randomly choose context and target points from the range $[0, t]$. We choose a random number of between 1 and 50 context and a fixed number of 100 target points, to produce a dataset. The linear interpolation is performed because otherwise the algorithm yields many more context and target points at regions where $R$ is large, however we would like the input locations of the data to be independent of the event rate $R$. Lastly we scale the target outputs by a factor of $10^2$ and add a positive constant of $10^{-2}$ to it. The reason for this last positive constant is that strictly speaking the marginal transformations used in the ConvGCNP models are not differentiable at 0, and we circumvent this pathology by adding this constant. We generate training data for 100 epochs, each consisting of 1024 iterations, each of which contains 16 different time series. For testing, we use a single epoch of 1024 iterations, each of 16 different time series.

**Neural architectures for the predator-prey tasks (ConvGNP, ConvNP, FullConvGNP):** We use the same model architectures for the ConvGNP, ConvNP and FullConvGNP in the predator-prey tasks as those which we used for the 1D synethtic Gaussian experiments, except we modify the discretisation resolution used by the various SetConv layers. More specifically, for the SetConvs of the ConvGNP, the ConvNP and that used in the mean parameterisation of the FullConvGNP, we use a discretisation of 16 points per time unit. For the SetConv used in the covariance architecture of the FullConvGNP we use 8 points per unit.

**Neural architectures for the ConvGCNPs:** For the ConvGCNPs we use identical architectures as for their ConvGNP counterparts, except we also compose the model with a marginal transformation. More specifically, we use a marginal transformation $\Phi_M^{-1}(\Phi_G(\cdot), \psi)$, where $\Phi_M$ is the CDF of the exponential distribution

$$\Phi_M(u) = 1 - e^{u/\psi},$$

where $\psi(\cdot) = \psi(\mathbf{x}_c, \mathbf{y}_c, \cdot)$ is an additional feature outputted by the ConvGNP. To avoid numerical instabilities during training, we limit $\psi$ to the range $[1, \infty)$, by passing the corresponding raw feature outputted by the ConvGNP through a SoftPlus function and adding 1 to the result.

**General notes:** We use the same number of features in the output layer of the ConvGNP models. The mean-field covariance uses two features, one for the mean and one for the marginal variance of the predictive, while the `linear` and `kvv` models both use the same number of $D_g = 32$ features for computing the predictive covariance. We use a learnable heteroscedastic noise variable for all models, and ReLU activation functions for all hidden layers. We optimise all models with Adam, using a learning rate of $5 \times 10^{-4}$. For the ConvNP we use 16 latent samples to evaluate the loss during training, and 512 samples during testing. We do not use any weight regularisation.

# E  ELECTROENCEPHALOGRAM EXPERIMENTS

**Details on datasets:** For the EEG experiments we use the publicly available EEG dataset, available on the UCI Datasets[4] website. We preprocess the data to remove invalid time series, leaving us with 7632 time series across 106 subjects. We pool all 106 subjects together (both control and alcoholic subjects) and sample 86 of the subjects for training, 10 for validation and 10 for testing. Each time series consists of 256 equispaced measurements over 64 EEG channels, from which we keep the seven channels with names `FZ`, `F1`, `F2`, `F3`, `F4`, `F5`, `F6`, following Bruinsma et al. (2020).

**Details on datasets for training the meta-models (ConvGNP, ConvNP):** We train each of our meta-models for 1000 epochs, each consisting of 256 iterations, at each of which the model is presented with a batch 16 different tasks. To generate each task, we first select a window size $W$ between 1 and 50. We then choose a window of size $W$ from the 256-long time series, and set the

---

[4]`https://kdd.ics.uci.edu/databases/eeg/eeg.data.html`

channels `FZ`, `F1`, `F2` within this window as context points. All other channels in this window, as well as all seven channels outside this window are used as context points.

**Details on datasets for testing the models (ConvGNP, ConvNP, MOGP):** During the testing phase of the meta-models, and the training phase of the MOGP, we use the same procedure for generating datasets as that used during the training phase of the meta-models, except we set the window size to a constant $W = 50$. We evaluate the models on the held-out test set. For the meta-models, we use 1 testing epoch of 256 iterations, each of which consists of presenting 16 tasks to the model. The MOGP model is trained and tested on individual time series without a meta-learning component, and is used as a non meta-learnt baseline to illustrate the benefits of meta-learning. We use 500 randomly sampled tasks, selected in the same way as the datasets used to test the meta-models. For each task, we fit the hyperparameters and the mixing matrix parameters on the context set of each dataset individually, and report the normalised predictive log-likelihood on the target set.

**Neural architectures for the EEG tasks (ConvGNP, ConvNP):** In these EEG experiments we use an on-the-grid ConvCNP architecture for the ConvGNP model, as specified in Gordon et al. (2020). More specifically, an on-the grid architecture does not require an input SetConv architecture, but instead uses a convolutional layer with positive-constrained weights. Following this convolution, a linear pointwise transformation is applied, followed by a UNet CNN, with six convolution and six transpose convolution layers. All layers use a kernel size of 5 and a stride of 2. The numbers of channels for the six convolution layers are

$$(c_{in}, c_{out}) = (16, 32), (32, 64), (64, 128), (128, 256), (256, 512), (512, 1024)$$

while for the transpose convolutions, the channels are

$$(c_{in}, c_{out}) = (1024, 512), (1024, 256), (512, 128), (256, 64), (128, 32), (64, 32)$$

followed by a last linear transformation mapping the output of a CNN to the dimensionality expected by the ConvGNP model. An output SetConv is used to interpolate the features to off-the-grid locations for purposes such as sampling. For the ConvNP, we follow the same approach as for the synthetic experiments, and stack two UNets with a latent variable layer in between. These are then followed by a SetConv for making off-the-grid predictions.

**General notes for the meta-models (ConvGNP, ConvNP):** We optimise both the ConvGNPs and the ConvNP using Adam and a learning rate of $2 \times 10^{-4}$. We use early stopping, selecting the model snapshot with the best performance on the validation set, through all of the 1000 training epochs, as the final model. In no case did we observe overfitting to the training set, even though we used no weight decay. In our experiments, we used $D_g = 512$ features for the `linear` and `kvv` models.

**Details of the MOGP model:** To demonstrate the benefits of the meta-learning approach, we also compare with a baseline multi-output Gaussian process (MOGP) model which, for every task separately, is trained from scratch, without a meta-learning component. To accelerate the training of many MOGP models, we use the approach by Bruinsma et al. (2020); an implementation is openly available on GitHub (link[5]). The MOGP models have $p = 7$ outputs and $m = 3$ latent processes with EQ kernels with initial length scales $10^{-2}$. All noises are initialised to $10^{-2}$. The mixing matrix and all hyperparameters are optimised using `scipy`'s implementation of L-BFGS-B.

## F  ENVIRONMENTAL EXPERIMENTS

### F.1  EXPERIMENTAL DESIGN

Experiments are conducted within the VALUE framework (Maraun et al., 2015) to facilitate comparison to existing downscaling methods. VALUE provides a suite of experiments for temperature and precipitation downscaling in an idealised framework mapping from two-degree resolution gridded reanalysis observations to station observations for daily maximum temperature at 86 locations around Europe. Models are trained on data from 1979-2003 and evaluated on data from 2003-2008. This mapping is then able to be applied to low-resolution climate model output to generate high resolution future projections for downstream tasks. Formally, we predict temperature $y$ at target (longitude, latitude) location $\mathbf{x}$ given low resolution predictors at two degrees resolution and orographic data at $\mathbf{x}$. We consider three experiments:

---

[5]`https://github.com/wesselb/oilmm`

1. Europe value only: models are trained on context data from ERA-Interim reanalysis from 1979-2002 and the 86 VALUE stations, and evaluated at these same locations for 2003-2008. The purpose of this experiment is to exactly reproduce the VALUE experiment protocol to facilitate comparison to state of the art baselines.

2. Europe all: In practice, a key limitation of current downscaling models is that they can only make predictions at a discrete set of locations determined at training time. Neural process models offer a significant advantage as the posterior stochastic process can be queried at an arbitrary location at inference time, regardless of the availability of training data. To evaluate how well the GNP models perform making predictions at new locations in the validation period we design a second experiment where models are trained on context data from ERA-Interim reanalysis from 1979-2002 and 3043 target stations around Europe, and evaluated on context data from 2003-2008 at the 86 VALUE stations.

3. Germany only: Germany has the highest density of stations of any country in Europe. As this is the scenario where modelling correlations between targets is most likely to improve performance we conduct a final experiment limiting the domain to Germany. Training is on ERA-Interim reanalysis context data from 1979-2002 and station data from 689 stations around Germany, with evaluation from 2003-2008 on 250 held out stations.

Locations of training and test stations for each experiment are shown in fig. 16.

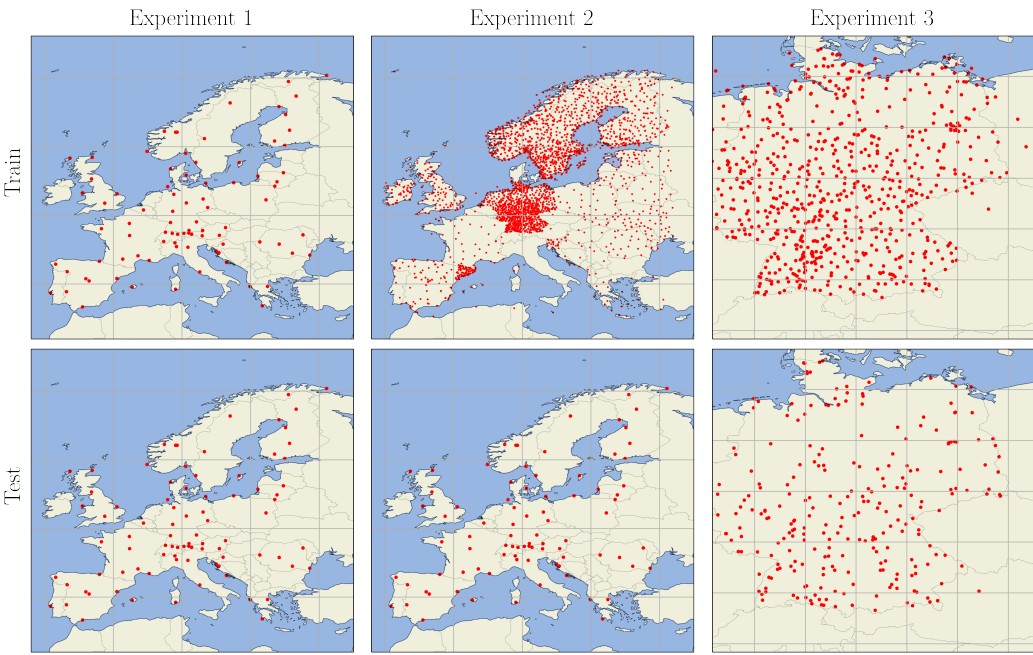

Figure 11: Locations of the training (top) and test (bottom) target locations for experiment 1 (left), experiment 2 (centre) and experiment 3 (right).

## F.2 DATA

Consistent with the VALUE protocol, we use the following context and target data:

**Context data** Context data are taken from ERA-Interim reanalysis (Dee et al., 2011) from 1979-2008 with daily temporal resolution and spatial resolution interpolated to two degrees using bilinear interpolation. We consider 25 variables:

- Surface: maximum temperature, mean temperature, northward and eastward wind.

- Upper atmosphere (850/700/500 hPa): specific humidity, temperature, northward and eastward wind.

- Invariant: angle of sub-grid scale orography, anisotropy of sub-grid scale orography, standard deviation of filtered subgrid orography, standard deviation of orography, geopotential, longitude, latitude and day of year transformed to $(\cos(t), \sin(t))$.

To account for sub-grid-scale topography, the context set also includes topographic variables at each target location. These include the true elevation, difference between reanalysis gridscale and true elevation and mTPI (Theobald et al., 2015).

**Target data** Target data are taken from weather station observations from 3129 weather stations around Europe reported as part of the European Climate Assessment Dataset (Klein Tank et al., 2002). This dataset includes daily observations of maximum temperature from 1979-2008.

### F.3 NEURAL ARCHITECTURES AND TRAINING

For the architectures we follow (Vaughan et al., 2021). The encoder consists of a CNN with six residual blocks, each consisting of two layers of depth-separable convolutions (Chollet, 2017) with a kernel size of 3 and 128 channels followed by ReLU activations. The encoder is followed by a SetConv layer with RBF kernel. Finally, a MLP is used to update predictions given the elevation of the target locations. This MLP consist of four hidden layers each with 64 units and ReLU activations. For the ConvGNP-linear and ConvGNP-kvv models we use $D_g = 128$ features for computing the covariance. For the ConvNP we use 32 channels for the latent function and 24 samples to calculate the neural process maximum likelihood at training time. Each model is trained for 500 epochs.

### F.4 ADDITIONAL SAMPLES

This section shows additional examples similar to fig. 9 comparing the ConvCNP and ConvGNP samples for different days.

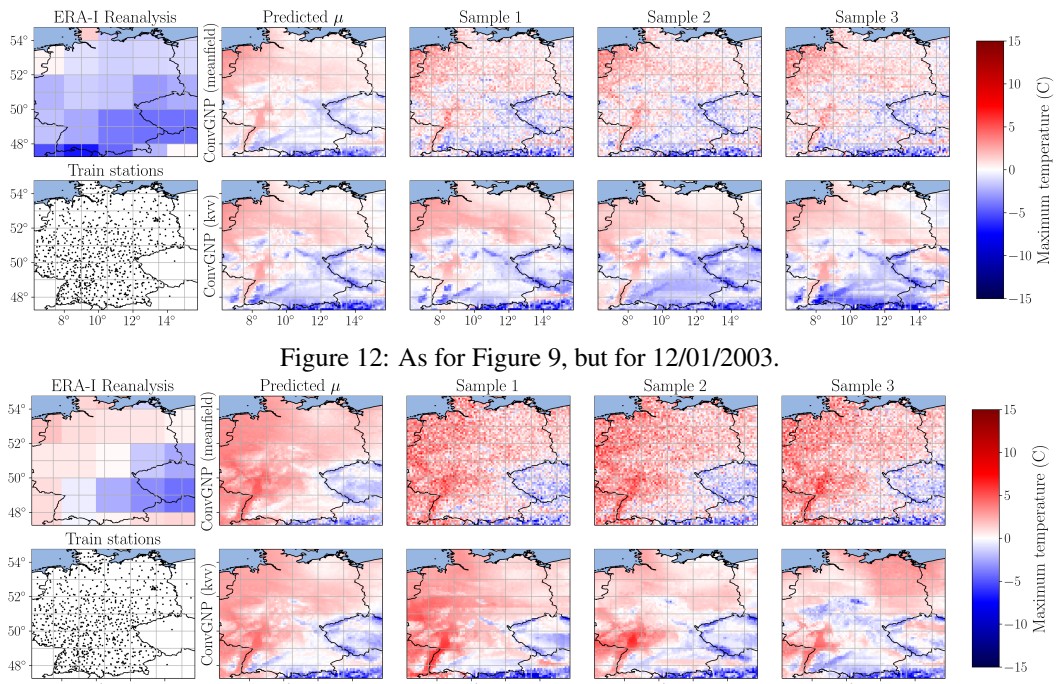

Figure 12: As for Figure 9, but for 12/01/2003.

Figure 13: As for Figure 9, but for 24/02/2003.

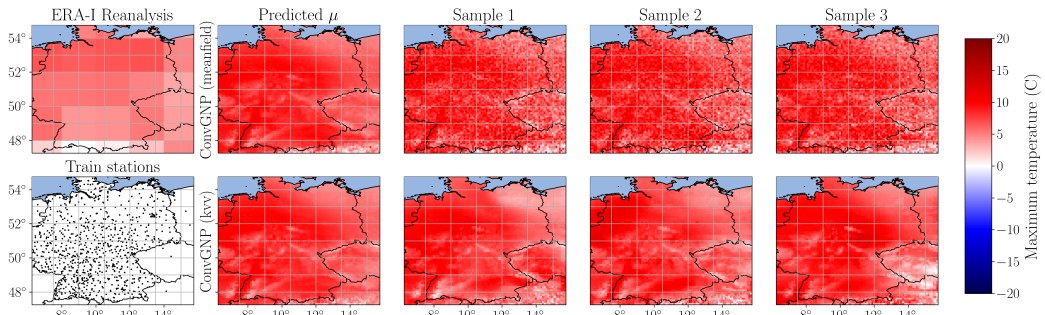

Figure 14: As for Figure 9, but for 06/03/2004.

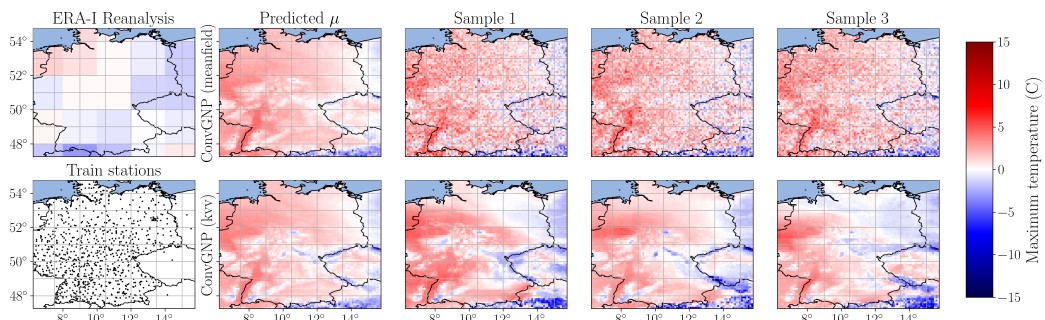

Figure 15: As for Figure 9, but for 23/01/2006.

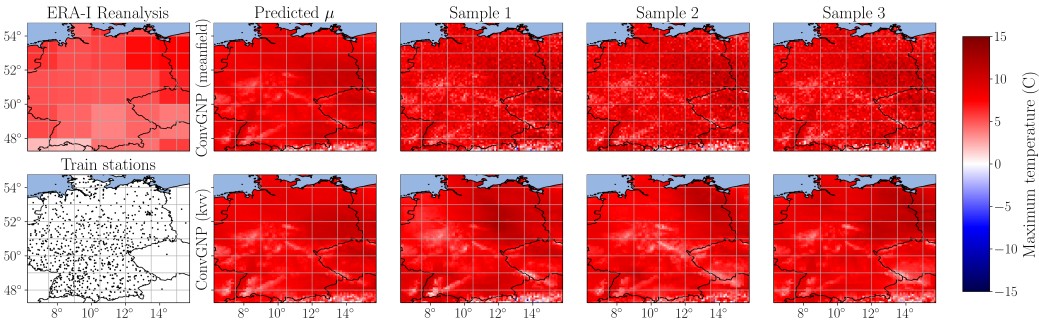

Figure 16: As for Figure 9, but for 04/04/2007.

