# OpenReview forum: "Practical Conditional Neural Process Via Tractable Dependent Predictions"
_ICLR.cc/2022/Conference — ICLR 2022 Poster_

### Official Review · Reviewer_Z7Bg · 2021-11-01

**Correctness:** 4
**Technical Novelty And Significance:** 2
**Empirical Novelty And Significance:** 2
**Recommendation:** 6
**Confidence:** 3

**Main Review:**

Strengths
1. The proposed methods are easy to understand and implement.
2.  The way of modeling dependencies in the input (or the output) through a covariance function looks simple but indeed improves the performance.
3. Using Copulae, the proposed methods can handle non-Gaussian distributions

Weaknesses
1. I think the novelty of the proposed methods is not striking. The proposed method looks like a direct implementation of a Gaussian process with deep kernel learning in the meta learning setting (through the form of NP): just including the global representation vector \mathbb{r} into the mean and the covariance functions for a GP.

2. The descriptions (in Introduction) related the reference [Bruinsma et al., 2021] looks a little bit confusing. First, the name “Gaussian Neural Process” (and the concept of modeling a Gaussian predictive using neural networks) is already introduced in the reference paper. Second, it sounds that the proposed methods are belonging to the same class as the FullConvGNP [Bruinsma et al., 2021] is (due to the same class name “Gaussian Neural Process”). Are the proposed methods also translation equivariant as the FullConvGNP is? If this is true, it is required to include proofs. If not, I think the introduction section should be carefully revised to clearly differentiate the proposed methods from the FullConvGNP [Bruinsma et al., 2021].


**Summary Of The Paper:**

The manuscript proposes variants of Neural process (NP), which can model correlation in the input (and in the output for multiouput regression). The main idea is to directly parameterize the mean and the covariance functions of a Gaussian predictive via neural networks. The authors also propose to use Copulae to handle non-Gaussian marginal distributions.

**Summary Of The Review:**

I like that fact that the authors include multiple data sets that can show the effectiveness of modeling dependencies in the input (or the output). However, I think the novelty of the proposed methods is not that impressive.

Minor comments

In the end of page 1: “CNPs model their respective outputs y_m and y_m’ independently, that is y_m \prep y_m’”. It would be better if the mathematical definition of the symbol “\prep” was included in the manuscript.

The current version of the manuscript is not printable (on Windows 10). I tried to print the manuscript out from Adobe reader and Chrome several times but failed to do it.

---

> ### Author Response · Authors · 2021-11-14
> **Response by the authors of Paper3542 to Reviewer Z7Bg**
>
> We thank Reviewer Z7Bg for their feedback. We are happy to hear that Z7Bg appreciates various aspects of our submission, including the points that our method is easy to implement and understand, that it achieves good performance on a variety of tasks, and that it can be extended to non-Gaussian and multivariate regression settings. We are also glad that the reviewer acknowledges the thoroughness of our experiments, including multivariate and large datasets, which demonstrate the empirical success of our method.
>
> **Summary of main issues and our responses**
>
> We now recap the main points of criticism of the reviewer and summarise our responses. We follow up with more detailed responses in subsequent comments.
>
> **Criticism 1: The novelty of the methods is not striking, they look like Deep Kernels**
>
> **Response 1:** Although both GNPs and DKs use neural networks in settings where GPs are often used, we stress that they have important differences. A DK uses a neural network and a kernel function to define a **GP prior**, which is conditioned on data using Bayes’ rule. By contrast, our approach uses neural networks to directly parameterise a **GP posterior**, without applying Bayes’ rule. Therefore, our approach has a complexity of $\mathcal{O}(N)$ in the number of conditioning points $N$, while DKs have complexity $\mathcal{O}(N^3)$. This distinction is especially important for scaling to large numbers of context points. We refer to this distinction in the **Related Work** section of our paper, and elaborate on it below.
>
> **Criticism 2: The description of the models and naming convention were not clear**
>
> **Response 2:** As we state in **Gaussian Neural Processes** in Section 2, the first model to parameterise the posterior covariance of a neural network
> > [...] was introduced by Bruinsma et al. (2021) [but is] challenging to scale to higher dimensions.
>
> The model of [Bruinsma et al., 2021] uses an expensive **fully convolutional architecture**, hence we refer to it as the FullConvGNP. However, this model is not the only way to parameterise a predictive covariance. We present an alternative for parameterising predictive covariances that is scalable and effective, giving rise to several new models. As we state in our paper, we collectively refer to all models with Gaussian predictives, including the FullConvGNP, as Gaussian Neural Processes (GNPs). Different architectures correspond to different members of this family (GNPs; AGNPs; ConvGNPs; FullConvGNPs), mirroring the naming convention for conditional neural processes (CNPs; ConvCNPs; etc). We explain this in **Neural architectures** in Section 2, but have amended our exposition in **Gaussian Neural Processes** and in **Neural architectures** to highlight this point. Further, we have added two more equations describing the ConvGNP and FullConvGNP models, which should help emphasise that they belong to the same family.
>
> **Criticism 3: It was not clear if the proposed methods are translation equivariant.**
>
> **Response 3:** From the models we introduced in this work, the GNP and AGNP are not equivariant, but the ConvGNP is translation equivariant just like the FullConvGNP. This is simple to show because the context set and target inputs enter the network through a DeepSet layer, which is translation equivariant, so the ConvGNP model inherits this property. **We will include a proof of this in the appendix of our submission (before the end of the review period on the 22nd of Nov.)** and reference it in the main text. We would also like to highlight that our method can be freely used with network architectures which may or may not be translation equivariant, as required by the problem at hand. This is an additional attractive feature of our approach, because it can be readily used with existing neural process architectures to build in appropriate inductive biases.
>
> **Empirical novelty and significance:** Lastly, we want to address the fact that the author’s criticism about the novelty of our contributions is
> > is not that impressive
>
> does not capture the significance of our empirical results. Our approach enables joint modelling and exact likelihood training on tasks to which the FullConvGNP would be prohibitively expensive to apply. In particular, our experiments on climate downscaling show that our method outperforms established baselines, including previous attempts using CNPs [Vaughan et al., 2021]. Further, our models enable practitioners to draw coherent function samples which are crucial for downstream estimation in risk mitigation and planning. This contribution bears its own scientific importance, and is impactful for the domain of climate modelling, and most probably also for other domains. To this end, we feel that the reviewer’s score for the **empirical novelty and significance** of our work does not fully capture the value of our contributions.
>
> In the thread below we address the reviewer’s comments about criticisms 1 and 2 in more detail.

---

> > ### Author Response · Authors · 2021-11-14
> > **Detailed response to the comments of Reviewer Z7Bg (3/3)**
> >
> > ### Minor comments
> >
> > **Symbol for independence of random variables:** Thank you for spotting that $\perp$ was not defined. This notation was intended to stress that $y_m$ is independent of $y_{m’}$. We have realised that this notation may have been confusing, so we have now removed it.
> >
> > **Printing on Windows 10:** We are sorry to hear you had issues with printing our manuscript. We are not sure what might be causing this issue for you, but we will reproduce and resolve it for the camera-ready version, so that this is not an issue for other readers. We hope this was not a major inconvenience for you.
> >
> > ### Summary
> >
> > We thank the anonymous reviewer for the time they took to review our submission. We were pleased that the reviewer appreciated the conceptual simplicity of our method, the fact that it brings computational and/or performance benefits over existing approaches and that it can accomodate handle multi-output and non-Gaussian regression. We hope that through our discussion we have highlighted the differences between our method and Deep Kernels. Importantly, our methods require a linear cost in the number of context points whereas Deep Kernels require a cubic cost, making our method much easier to scale to larger datasets. We further hope that our discussion on the naming convention we followed for the models has clarified the nomenclature we have used. We largely followed the naming convention of the existing Neural Process literature. We also hope to have clarified the reviewer’s question on equivariances, and hope that the addition of a straightforward proof in our manuscript will address their initial concern. In addition we hope that after this discussion the reviewer appreciates the novelty of our work. We believe the conceptual simplicity of our method adds to, rather than subtracts from, our work, and makes it more likely that our method is adopted by practitioners in a range of domains.

---

> > ### Author Response · Authors · 2021-11-14
> > **Detailed response to the comments of Reviewer Z7Bg (2/3)**
> >
> > ### Criticism 2: The description of the models and naming convention were not clear
> >
> > The reviewer commented that
> > > The description (in Introduction) related to the reference [Bruinsma et al., 2021] looks a little bit confusing.
> >
> > In particular, the reviewer makes the point that
> > > The name Gaussian Neural Process (and the concept of modelling a Gaussian predictive using neural networks) is already introduced in the reference paper.
> >
> > and also that
> > > It sounds that the proposed methods belong to the same class as the FullConvGNP [Bruinsma et al., 2021].
> >
> > **2.1 Gaussian Neural Processes:** We would like to emphasise that, as the reviewer remarked, our paper is not the first to propose modelling predictive covariances using neural networks. We clearly state this in the paragraph **Gaussian Neural Processes** in Section 2, pointing out that the first such model was introduced by Bruinsma et al., 2021. However, the method of Bruinsma et al. is not the only way to parameterise a predictive covariance using a neural network. Therefore, we find it sensible to refer to all models which parameterise Gaussian predictions with the term “Gaussian Neural Processes”. This term includes the model of Bruinsma et al. and all the models presented in this work.
> >
> > **2.2 Naming convention for the members of the GNP family:** The model of Bruinsma et al. uses a fully convolutional architecture to parametrise the predictive covariance, hence we refer to it as a Fully Convolutional Gaussian Neural Process (FullConvGNP). This model involves highly expensive 2D-dimensional convolutions which make it difficult to scale to higher dimensions and impactful problems. By contrast, the method we propose in this work relies on using feature vectors to parameterise a predictive covariance, instead of a fully convolutional architecture. Our method can be readily used with arbitrary neural architectures according to the task at hand. For the names of these models, we follow the convention in the CNP literature, which adds a modifier to the model’s name according to the architecture used. In particular, a CNP using a feedforward DeepSet is simply referred to as a CNP, hence we refer to a Gaussian Neural Process with a feedforward DeepSet as a GNP. Similarly we refer to GNPs using attentive or convolutional architectures as AGNPs and ConvGNPs respectively. Note that our ConvGNP relies on a convolutional architecture, which, however, still uses feature vectors to avoid the 2D-dimensional convolutions of the FullConvGNP. We have modified the paragraph **Neural architetctures** to clarify this naming convention. We also included two equations describing the FullConvGNP and ConvGNP models, which should help clarify the fact that they both belong to the GNP family, though they involve different architectures which bring about large different computational and memory costs.

---

> > ### Author Response · Authors · 2021-11-14
> > **Detailed response to the comments of Reviewer Z7Bg (1/3)**
> >
> > ### Criticism 1: The novelty of the methods is not striking, they look like Deep Kernels
> >
> > The reviewer suggested that
> > > The novelty of the proposed methods is not striking.
> >
> > and commented that
> > > The proposed method looks like a direct implementation of a Gaussian Process with deep kernel learning.
> >
> > We would like to stress that while GNPs appear similar to Deep Kernels (DK) [Patacchiola et al., 2019; Calandra et al., 2016; Wilson et al., 2015], the two model classes differ in an important way.
> >
> > **1.1 DKs parameterise a prior:** DKs build on GPs by mapping the input variable $x$ through a neural network network and passing the resulting feature to a covariance function $k$, to define a **parametric GP prior** as $f \sim \mathcal{GP}(0, k_{DK}(x, x’))$, where $k_{DK}(x, x’) = k(g_{DK}(x), g_{DK}(x’))$ and $g_{DK}$ is a neural network. In order to make predictions, a DK model applies **Bayes’ rule** to condition on the observed data and make predictions, via the formulae of the predictive mean $\mathbf{m} = k(\mathbf{x}_t, \mathbf{x}_c) [k(\mathbf{x}_c, \mathbf{x}_c)]^{-1} \mathbf{y}_c$ and predictive covariance $\mathbf{K} = k(\mathbf{x}_t, \mathbf{x}_t) - k(\mathbf{x}_t, \mathbf{x}_c) [k(\mathbf{x}_c, \mathbf{x}_c)]^{-1} k(\mathbf{x}_c, \mathbf{x}_t)$. Therefore, the computational complexity of conditioning on $N$ context points is $\mathcal{O}(N^3)$ due to the matrix inversion required.
> >
> > **1.2 GNPs parameterise a predictive:** By contrast, GNPs, just like all other members of the neural process family, enjoy a much lower $\mathcal{O}(N)$ complexity. This is because, unlike DKs which define a prior covariance, GNPs directly output a **predictive covariance** instead. They achieve this by using flexible neural networks, which operate on set inputs [Zaheer et al. 2017, Gordon et al. 2020] to learn a mapping from a conditioning set $(x_c, y_c)$ to **predictive covariance functions**. More specifically, our models map the context set to a latent representation $\mathbf{r} = r(x_c, y_c)$, which is then passed to another network which outputs a feature vector $g_i = g_{GNP}(x_{t, i}, \mathbf{r})$. Lastly, this feature vector is passed to an appropriately chosen positive definite function $k$ to produce a predictive covariance $\mathbf{K} = k(g_i, g_j)$. The cost of the forward step of our neural networks is linear in the number of context points, whereas the cost of DK is cubic. This has significant implications for the practical use of Neural Process models, since they can be scaled to large datasets with ease. We acknowledge that certain of the equations which appear in both classes of models have a similar form, which might confuse readers. **We highlighted the distinction between GNPs and DKs in the Related Work section, however we will rephrase that paragraph (before the end of the discussion period on the 22nd of Nov.) to better highlight the differences between the models, and reference this in the background section.**
> >
> > **1.3 CNPs are not DKs:** We would also like to bring to the reviewer’s attention that the entire CNP family (including all their architectural variants like the ConvCNP, GroupCNP, as well as the GNP models which we present here and the FullConvGNP) operate in the fashion described above: they parameterise a predictive distributions directly, thereby avoiding high computational costs at test time, from which DKs suffer. These models are not a direct implementation of a GP with a Deep Kernel, and should not be conflated with DKs.

---

> > ### Author Response · Authors · 2021-11-22
> > **Revision of manuscript**
> >
> > We would like to bring to the reviewer’s attention the recent changes we have made to our manuscript to address their points of criticism. In particular, we have added further equations (eqs. 5 and 13) to highlight the differences between the ConvGNP and FullConvGNP, to clarify the novelty of our contributions. We have also modified the **Deep Kernels** paragraph in the Related Work section to stress the differences of our approach to DKs. In addition, we added two appendices: appendix A includes the proof for the equivariance of the ConvGNP model, as requested by the reviewer; and appendix B includes a more detailed benchmarking including runtime and memory footprint information for the various models, which should help highlight the computational benefits of the ConvGNP model over the FullConvGNP, and clarify the significance of our contributions.
> >
> > We believe that our changes address the points of criticism raised by the reviewer, and hope that the reviewer considers increasing their overall score.

---

### Official Review · Reviewer_WnVi · 2021-11-02

**Correctness:** 4
**Technical Novelty And Significance:** 3
**Empirical Novelty And Significance:** 3
**Recommendation:** 6
**Confidence:** 4

**Main Review:**


When I saw 'Tractable Dependent Predictions' in the title, I assumed that a normalizing flow was being used to capture the joint distribution. This is a modern, flexible family of density estimators for which computing the likelihood is tractable. I was disappointed when I found that these weren't used in the paper. The copula model is very close to a full normalizing flow model. Can you explain what would be necessary to extend your method to a full normalizing flow?

I found the discussion of fullconvgp inadequate, as it seems like the most relevant baseline for capturing these dependencies. It would be very helpful if you included an equation or two. I shouldn't need to dig into the literature to understand the difference between your method and prior work. What exactly is the difference between fullconvgp and convgnp?

I was disappointed that there were no error bars. What sources of variance are you averaging over?

Can you make the analysis of fig 3 more quantitative? Right now, the assertion that the samples are better is too qualitative. Can you check, for example, a q-q plot at a few x axis locations?

I'm curious how your model behaves when the data truly has diagonal covariance. I expect that there are off-diagonal artifacts due to overfitting. Can you run a quick experiment to check this?





**Summary Of The Paper:**

There is a long line of recent interesting work on neural processes, a scalable and more flexible alternative to GPs for performing prediction at a set of test points (x1, ..., xm) given a conditioning set ((x, y)_1, ..., (x, y)_n). This mapping is learned via meta-learning.

This paper addresses a core issue of the popular conditional neural process: the predictions at each test point are conditionally independent given the conditioning set. This is an inappropriate modeling assumption for many real-world datasets. In response, the authors propose to go beyond a non-diagonal Gaussian to describe the joint distribution. For example, they use some structured Gaussian covariances (linear, kvv) and also a Gaussian copula model.

They demonstrate both qualitatively and quantitatively that modeling these dependencies improves the performance of the model on a variety of datasets spanning application domains.




**Summary Of The Review:**

I feel that the contribution is fairly incremental and I'm disappointed that they did not consider full normalizing flow models. However, I found the exposition engaging and the experiments thorough.

---

> ### Author Response · Authors · 2021-11-14
> **Response by the authors of Paper3542 to Reviewer WnVi**
>
> We thank Reviewer WnVi for their feedback. We are glad to hear the reviewer finds this line of work on Neural Processes interesting, and **appreciates that our work improves on existing Neural Process models on a variety of settings**. We were very happy that the reviewer was pleased with the **quality of exposition and the thoroughness of our experiments**. We believe that these, experimentally demonstrated and sizeable, performance improvements, together with the simplicity of our approach and the quality of exposition, are key contributions made in this work. We would also like to address the reviewer’s main criticisms of the submission.
>
> Summary of main issues and our responses
>
> We now recap the main points of criticism of the reviewer and summarise our responses. We follow up with more detailed responses in subsequent comments.
>
> Criticism 1: Normalising flows could be used to model joint dependencies.
>
> Response 1: The only way to compose an invertible transform (like an NF) and a CNP is to use a transformation of the marginals; otherwise, such a CNP+NF model is inconsistent (it is not a valid stochastic process). We investigate marginal transformations in Section 3 in the paper and elaborate on this point in **CNPs and NFs** below.
>
> Criticism 2: The discussion of the FullConvGNP [Bruinsma et al., 2021] is inadequate.
>
> Response 2: In the latest revision of the submission, we have included additional equations and accompanying text, to better explain the differences between the two models. We also include a more detailed discussion in **Discussion of the FullConvGNP** below.
>
> Criticism 3: The contributions are too incremental.
>
> Response 3: We consider the simplicity of the proposed method a feature rather than a downside. In particular, a key contribution of the submission is demonstrating that our method, while simple, outperforms established baselines on a large-scale climate downscaling task. This particular task is of current interest to climate scientists [Vaughan et al., 2021], to whom modelling output dependencies is highly valuable. We believe the simplicity and effectiveness of our method can encourage domain experts, such as climate scientists, to adopt it in their work.
>
> Finally, we note that the reviewer scored the empirical novelty and significance of the submission as “Not Applicable”. Whilst this may have been an accidental omission on the reviewer’s part, we believe that the empirical contribution of our submission is substantial. If this mark was accidentally omitted, we would like to invite the reviewer to reconsider the mark they provided for this aspect of our submission.
>
> In the thread below, we address the criticisms of the reviewer in more detail.

---

> > ### Author Response · Authors · 2021-11-14
> > **Detailed response to the comments of Reviewer WnVi (5/5)**
> >
> > ### Summary
> >
> > We thank the anonymous reviewer for the time they took to review our submission. We are very happy that the reviewer appreciated the experimental thoroughness of our paper and our quality of exposition. We hope that through reading our elaboration on NFs, the reviewer will appreciate that we gave the CNP+NF model class some serious thought, but due to theoretical limitations, we could not find a way to make it work for our setting. We also hope to have further clarified the differences of the ConvGNP and the FullConvGNP model. Lastly, we hope that our discussion on the significance of our contributions has made it clearer that our approach, while simple to understand, is more than incremental. In particular, it provides substantial improvements in performance and/or scalability over existing models in several different tasks, including an impactful climate downscaling task. We hope this discussion has addressed the reviewer’s concerns, in which case we would ask them to consider improving the score of our submission. We look forward to hearing their response and any further questions they may have.

---

> > > ### Comment · Reviewer_WnVi · 2021-11-19
> > > **Raising my rating.**
> > >
> > > I appreciate your very thorough response. In particular, thanks for the clarifications around normalizing flows. This is not obvious, and would definitely be a valuable part of the camera-ready paper. If you make the various updates to the paper that you suggest regarding this, performance metrics, FullConvGP, etc., then I think it will be well received by the ICLR community.

---

> > > > ### Author Response · Authors · 2021-11-22
> > > > **Revision of manuscript**
> > > >
> > > > We thank the reviewer for their response to our comments. We are glad the reviewer found our response useful. We appreciate that the reviewer raised their scores for technical and empirical novelty and significance.
> > > >
> > > > We would like to bring to the reviewer’s attention the changes we have made to our manuscript, especially related to the above discussion. Among other changes, we have included a discussion on Normalising Flows (Appendix A.2), and make reference to this discussion and to the relation with Normalising Flows in Section 3, following the suggestion of the reviewer. We have also added further equations which highlight the differences between the ConvGNP and FullConvGNP, and a section (appendix B) benchmarking and explaining the performance differences between the various models in terms of runtime and memory. We have modified the statements in our plots to stress the fact that the error bars are too small to be seen in the scale of our plots.
> > > >
> > > > We hope the reviewer finds these changes useful and will consider raising the overall score too.

---

> > ### Author Response · Authors · 2021-11-14
> > **Detailed response to the comments of Reviewer WnVi (4/5)**
> >
> > ### Further comments
> >
> > We now address some further points raised by the reviewer.
> >
> > **Sources of errors and error bars**
> >
> > The reviewer commented on the presentation of the errors in our plots, saying
> > > I was disappointed that there were no error bars.
> >
> > and also asked
> > > What sources of errors are you averaging over?
> >
> > For all experiments in our paper, we train each of the models once and evaluate them over a large number of datasets. Therefore there is variance in the predictive log likelihood due to the different datasets that we evaluate on. Because we use a large number of datasets (e.g. figure 4 involves $\approx 1.6 \times 10^4$ datasets) to compute an average log-likelihood, **the errors are negligible and too small to be seen in the scale of our plots**. Each of our plots already includes an explanatory phrase which highlights this point, stating
> >
> > > **Figure 3: [...] Errors too small to be seen here.**
> >
> > > **Figure 4: [...] Errors too small to be seen here.**
> >
> > > **Figure 5: [...]  Errors for the per-datapoint predictive log-likelihoods and threshold estimation task log-likelihoods (right) are too small to be seen.**
> >
> > Therefore, the reason why the reviewer did not see any error bars is because we average over sufficiently many datasets when evaluating the models, that the error bars are too small for the scale of the plots. **We will however modify these statements to stress this fact (before the end of the discussion period on the 22nd of Nov.)..**
> >
> > **Quantitative analysis of samples**
> >
> > The reviewer commented that
> > > The assertion that the samples [in Figure 3] are better is too qualitative.
> >
> > We assume that the reviewer refers to the claim made at the end of Section 4.1, under the paragraph titled **Predictive samples**, where we claimed that
> > > The ConvGNP is the only conditional model (other than the FullConvGNP) which produces high-quality posterior samples.
> >
> > We assume the reviewer refers to this claim, because at no other point in the paper do we make reference to the quality of the samples shown in Figure 3.
> >
> > **Rephrasing our claim:** We agree that in the current statement of this claim we do not explain how we judge sample quality. This claim could thus be misleading and should be addressed. We thank the reviewer for pointing out this issue. We can amend this claim by referring to the results in Figure 4. This figure shows that the ConvGNP (kvv) and FullConvGNP models attain the best predictive log-likelihood out of the models we tested. Below, we argue that the predictive log-likelihood (averaged over a large number of datasets) is a sufficient and objective metric to decide which models have best sample quality.
> >
> > **Predictive log-likelihood is a sufficient and objective metric for judging samples:** The sample quality depends on the accuracy of the prediction map $\pi$, and in particular how well it approximates the ground truth prediction map, which we denote $\pi^*$ here. Given prediction maps $\pi_1$ and $\pi_2$, an objective metric to ascertain which has the best sample quality, is the **KL divergence** between the models the ground truth prediction map, i.e. $KL(\pi^*(y_t; x_c, y_c, x_t) || \pi_i(y_t; x_c, y_c, x_t))$ for $i = 1, 2$. The prediction map with lowest KL will be closer to the ground truth, and will produce the most faithful samples. $KL = 0$ signifies a perfect prediction map, when the model predictive is equal to the ground truth, and the larger the KL, the worse the predictive and thus also the samples.
> >
> > The KL can be written as $KL(\pi^*(y_t) || \pi_i(y_t)) = \int \pi^*(y_t) \log \pi^*(y_t) dy_t - \int \pi^*(y_t)\log \pi_i(y_t) dy_t$, where we have abbreviated $\pi(y_t; x_c, y_c, x_t) = \pi(y_t)$ for compactness. Since $\int \pi^*(y_t) \log \pi^*(y_t) dy_t$ depends on the ground truth and not the model $\pi_i$, the model which minimises the KL is that which maximises $\int \pi^*(y_t) \log \pi_i(y_t) dy_t$. This quantity is the predictive log-likelihood, and **the empirical predictive log-likelihood shown in Figure 4 is an unbiased estimator of this quantity**. Therefore, averaging over many datasets, we can compare the empirical average log-likelihoods of the models in Figure 4 to decide which has the best samples (higher log-likelihood means better samples), in an objective way. From Figure 4 we see that the best models (and thus the ones which produce the best samples) are the ConvGNP (kvv) and FullConvGNP. **We appreciate that we did not make reference to this point, but will amend the text to reflect it (before the end of the discussion period on the 22nd of Nov.), making the metric for sample quality explicit.**
> >
> > **Additional experiment with mean-field data:** We agree that this would be an informative experiment to run using our models. We are currently working on producing the required results and will either add these either in the appendix or share them publicly here using an anonymised file sharing system (before the end of the discussion period on the 22nd of Nov.).

---

> > ### Author Response · Authors · 2021-11-14
> > **Detailed response to the comments of Reviewer WnVi (3/5)**
> >
> > ### Criticism 3: The contributions are too incremental.
> >
> > The reviewer suggested that
> > > The approach is fairly incremental.
> >
> > We believe that two reasons why the reviewer may have formed this impression are that (1) initially, the differences between the ConvGNP and the FullConvGNP was not fully clear; and (2) our contributions are simple to describe and understand. We address each of these points below.
> >
> > **3.1 Clarifying the novelty of our method over the FullConvGNP:** We believe that one likely reason why the reviewer thought our approach is incremental, is that the differences between the ConvGNP and FullConvGNP were not fully clear initially. We hope that the earlier discussion on the two models (response section 2) has clarified these differences. We also hope these comments have helped the reviewer appreciate that our approach is substantially more computationally and memory efficient than the FullConvGNP of Bruinsma et al. [2021]. Although conceptually simple, our approach allows us to tackle impactful applications, such as climate downscaling, which were not previously possible with this type of model. To further stress the computational and memory benefits of our approach over the FullConvGNP, **we will add a table of time and memory-usage measurements (before the end of the discussion period on the 22nd of Nov.), highlighting the benefits of our method, to our paper**. We note that the proposed models are, to the best of our knowledge, the only neural processes which can model output dependencies, are trainable via an exact log-likelihood, and can be scaled to practical problems in higher dimensions. Also, as explained in response section 2.3, our approach does not necessarily require convolutional architectures at all, which could be appropriate for a wide range of real tasks.
> >
> > **3.2 Conceptual simplicity of our contributions:** Further, another reason which might have caused the reviewer to think our approach is incremental, is that it is simple to understand. However, we consider the conceptual simplicity of our approach to be a merit rather than a flaw. In particular, the approach is easy to understand and implement, whilst significantly improves performance over alternative methods, and is computationally affordable. We expect that these features make our method appealing to, and easy to adopt by, domain experts in other fields. In climate downscaling for example, we have demonstrated that our method outperforms established baselines [Vaughan et al., 2021], demonstrating a clear benefit for this line of research, and we believe that other domains could directly benefit from using our method.
> >
> > **3.3 Significance of empirical results:** We also want to stress that a significant part of our contributions is that we demonstrate the empirical success of our methods on a variety of tasks. As the reviewer points out
> > > [...] the experiments [were] thorough
> >
> > and included tasks with both real and synthetic data, non-Gaussian and multivariate regression as well as an impactful application on climate modelling, which bears its own scientific value. Especially on the latter, the fact that our models outperform established climate downscaling baselines is an important contribution which, as we mentioned in 3.3 above, has direct impact on this domain, and can yield further modelling benefits in a broad range of other domains.

---

> > ### Author Response · Authors · 2021-11-14
> > **Detailed response to the comments of Reviewer WnVi (2/5)**
> >
> > ### Criticism 2: The discussion of the FullConvGNP [Bruinsma et al., 2021] is inadequate.
> >
> > The reviewer commented that
> > > [They] found the discussion of the FullConvGNP inadequate.
> >
> > and suggested that
> > > [...] it would be very helpful if [we] added an equation or two.
> >
> > Further, the reviewer asked us to clarify
> > > What exactly is the difference of the FullConvGNP and the ConvGNP?
> >
> > We’d like to thank the reviewer for pointing out that our exposition could be improved using additional equations. **We have added two equations in section 2**, one describing the FullConvGNP (1.1 below) and another describing the ConvGNP (1.2 below), so that the two can be easily compared, and the novelty of our approach is made clearer. We also discuss the difference between the two models below.
> >
> > **2.1 Equation on the FullConvGNP:** The first equation describes how the FullConvGNP computes its covariance matrix $\mathbf{K}$ and takes the form (eq. 13 in our revision)
> >
> > $(\mathbf{x_c}, \mathbf{y_c}) \rightarrow (\mathbf{\tilde{x}}, \mathbf{h})  \rightarrow \mathbf{r} = PSD(CNN(\mathbf{h}))  \rightarrow \mathbf{K_{ij}} = \sum^L_{l = 1} \psi(x_{t, i}, \tilde{x_l}) r_l \psi(\tilde{x_l}, x_{t, j}),$
> >
> > where in the first step, $(\mathbf{x}_c, \mathbf{y}_c)$ is mapped to a grid of values $\mathbf{h}$ on a $2D$-dimensional grid at locations $\mathbf{\tilde{x}} = (\tilde{x}_1, \dots, \tilde{x}_L)$, where $\tilde{x}_l \in \mathbb{R}^{2D}$, using a SetConv layer [Gordon et al. 2020], in the second step $\mathbf{h}$ is passed through a CNN with $2D$-dimensional convolutions to produce values $\mathbf{r}$ on the same grid, followed by a PSD transformation which forces the $\mathbf{r}$ tensor to be positive definite, and in the third step $\mathbf{r}$ is aggregated using RBFs $\psi$. **Crucially, this parameterisation of $\mathbf{K}$ requires highly expensive $2D$-dimensional convolutions, which become extremely expensive for $D > 1$, both in terms of computation and memory.**
> >
> > **2.2 Equation on the ConvGNP:** The second equation describes how the ConvGNP parameterises its covariance. Instead of parameterising $\mathbf{K}$ directly, the ConvGNP produces a feature vector $g \in \mathbb{R}^{D_g}$ for each target point, and passes this through an appropriately chosen positive-definite function $k$ to parameterise a posterior covariance. The feature vector $g$ for target point $x_{t, i}$ is computed according to
> >
> > $(\mathbf{x_c}, \mathbf{y_c}) \rightarrow (\mathbf{\tilde{x}}, \mathbf{h})  \rightarrow \mathbf{r} = CNN(\mathbf{h})  \rightarrow g(x_{t, i}, \mathbf{r}) = \sum^L_{l = 1} \psi(x_{t, i}, \tilde{x}_l)~r_l,$
> >
> > where, crucially, $(\mathbf{x}_c, \mathbf{y}_c)$ is mapped to a grid of values $\mathbf{h}$ on a **$D$-dimensional grid at locations $\mathbf{\tilde{x}} = (\tilde{x}_1, \dots, \tilde{x}_L)$, where $\tilde{x}_l \in \mathbb{R}^{D}$** and the CNN involves $D$-dimensional convolutions. The lower-dimensional convolutions make the ConvGNP **substantially cheaper in terms of compute and memory, and make it possible to scale to impactful problems such as the climate application which we present**.
> >
> > **2.3 Parameterising the covariance with feature vectors:** Further, using our approach it is not necessary to use convolutions to parameterise the covariance at all. More specifically, any DeepSet architecture [Zaheer et al., 2017] (not necessarily convolutional) can be used to produce features which we can use to parameterise the covariance. As we explain in Section 2 of our submission, $g(x_{ti}, \mathbf{r})$ and $\mathbf{r} = r(\mathbf{x}_c, \mathbf{x}_t)$ can be parameterised using a simple feedforward network and a DeepSet respectively, and then fed to an appropriately chosen PSD function $k$ to express the predictive covariance. Therefore, our approach goes beyond convolutions, permitting us to use any, e.g. feedforward or attentive, network. The GNP and AGNP models in our paper are such examples. These models could be especially useful for tackling datasets where convolutions are not an appropriate inductive bias.
> >
> > We hope that this clarifies the reviewer’s query, and also **highlights the novelty of our approach**. Instead of representing the covariance function directly, the ConvGNP learns to produce feature vectors which are then used to parameterise the covariance. Our approach is not limited to convolutional architectures and can readily be used with virtually any neural process architecture.

---

> > ### Author Response · Authors · 2021-11-14
> > **Detailed response to the comments of Reviewer WnVi (1/5)**
> >
> > ### Criticism 1: Normalising flows could be used to model joint dependencies.
> >
> > The reviewer commented that
> > > [The authors] did not consider full normalising flow models.
> >
> > and would have liked to see a normalising flows used
> > > [...] to capture the joint distribution.
> >
> > We considered combining NFs and CNPs in the start of this project. What we found is that composing a CNP with a NF to model joint dependencies will not give a consistent model in general; and the only NFs which give consistent models for all CNPs are non-interacting transformations of the marginals, which cannot model joint dependencies. Although, for a particular CNP, there can exist a NF which gives a consistent model, constraining a NF in this way is a hard research problem and beyond the scope of what we investigate in this paper. This answers the reviewer's question
> > > Can you explain what is necessary to extend your method to a full normalising flow?
> >
> > Namely, to extend our method to normalising flows, the problem of constraining normalising flows to ensure that the composition of a NF and a CNP is a consistent model will need to be solved.
> >
> > The subsequent paragraphs discuss the subtleties of combining CNPs and NFs in more detail.
> >
> > **1.1 Composing CNPs and NFs:** We originally considered composing an arbitrary invertible map $\Theta$ with a mean-field CNP. As WnVi suggests, $\Theta$ could model joint dependencies in the output as well as non-Gaussian marginals. In particular, suppose we have a mean-field CNP with prediction map $\pi(\dot~; x_c, y_c, x_t)$, and want to apply an invertible map $\Theta$ to the output of the CNP. In other words, to draw a sample from the predictive distribution, we first draw $(u_{t1}, \dots, u_{tM}) \sim \pi_G(u_t; x_c, y_c, x_t)$, and then compute $(y_{t1}, \dots, y_{tM}) = \Theta(u_{t1}, \dots, u_{tM})$. If we want to retain a model which is consistent under marginalisations of the predictive, this poses **theoretical limitations** on $\Theta$.
> >
> > **1.2 Theoretical limitations:** First, $\Theta$ must be able to handle variable-length tuples as arguments, to handle arbitrary target points. Thus, we should be able to query $\Theta(u_{t1}, \dots, u_{tM})$ with any number $M$ of inputs. Second, the model should be consistent under marginalisations: making a joint prediction for target variables $y_{t1}$ and $y_{t2}$, and then marginalising over $y_{t2}$, should be equivalent to making a prediction over $y_{t, 1}$ directly. To be clear, consider drawing a sample $(u_{t1}, u_{t2}) \sim \pi_G(u_t; x_c, y_c, x_t)$ and then passing this through $\Theta$ to obtain $(y_{t, 1}, y_{t, 2}) = \Theta(u_{t1}, u_{t, 2})$. Marginalisation consistency requires that the distribution of $(\Theta(u\_{t1}, u\_{t2}))\_i$ is equal to that of $\Theta(u_{ti})$, which, for a careful choice of $\Theta$ and distribution of $(u_{t1}, u_{t2})$ can be true, but in general will not be true. If we want to construct a NF $\Theta$ such that it gives a consistent model for _all_ underlying CNPs, then that requires the the stronger condition that $(\Theta(u\_{t1}, u\_{t2}))\_i = \Theta(u_{ti})$ almost surely. From this condition, we see that $\Theta(u_{t1}, u_{t2}) = (\Theta(u_{t1}), \Theta(u_{t2}))$, i.e. $\Theta$ can only be a **marginal transformation**.
> >
> > **1.3 A CNP+NF model cannot model dependencies:** Now, note that if $\pi$ is mean-field and $\Theta$ is a marginal transformation, then it follows that the $y_t$ variables are independent. Therefore, it is not possible to compose a mean-field CNP with an NF to model joint dependencies in this way. Instead, we must rely on $\pi$ for modelling dependencies, and $\Theta$ for learning the marginals, which is what we did in the paper (see Sections 3 and 4.2)
> >
> > **1.4 Copulae and Normalising Flows:** In Section 3 we considered a structured form for $\Theta$, and we highlight its connection to Copula Processes. As WnVi points out, this model relates to NFs. In fact, these Copulae can be thought of as a particular type of NF in 1D, and vice versa. Originally, we did not highlight this connection, because typically NFs are applied to multi-dimensional variables while the marginals are 1D distributions (though examples of 1D flows have been explored before, see [Neural Spline Flows (NSF)](https://arxiv.org/pdf/1906.04032.pdf)). While this is a naming choice on our part, we have realised we should make the connection to flows explicit, after the reviewer’s comment. **We will revise our manuscript (before the end of the discussion period on the 22nd of Nov.) to point out the relation to NFs in Section 3**, including pointers to relevant literature such as the NSF. Last, note that while we could use an arbitrary invertible map $\Theta$, in our experiments the data marginals were simple enough and our exponential parameterisation worked well. We did not find it necessary to explore more complicated $\Theta$’s though this is entirely possible under our framework.

---

### Official Review · Reviewer_g5Qp · 2021-11-03

**Correctness:** 4
**Technical Novelty And Significance:** 2
**Empirical Novelty And Significance:** 3
**Recommendation:** 8
**Confidence:** 4

**Main Review:**

The authors introduce a relatively straightforward extension for Gaussian neural processes in which both mean and (half of the) covariance functions are specified as neural networks, and the covariance function is either explicitly calculated as an inner product in (7) or as a squared exponential covariance function modulated in magnitude by an auxiliary neural network and calculated using the outputs of a neural network rather than the input data itself (x_t, x_c, y_c). Their multi-output and non-Gaussian strategy follows the modulated kernel and Gaussian copula formulation, respectively. Comprehensive experiments on both artificial and real data demonstrate the advantages of the proposed model over the related, but more computationally expensive, fullConvGNP model. Moreover, on real data, the proposed model outperforms both mean field, convNP and MOGP approximations.

Something that is not discussed in the paper is the setting of the length scale of the squared exponential kernel.

Considering that one of the motivations of the proposed approach is how prohibitive existing approaches are, having estimates of computational cost and/or runtime experiments could be a welcome addition to the paper.

**Summary Of The Paper:**

The authors present a class of neural process models that are able to produce correlated predictions while amenable to exact, simple and scalable maximum likelihood optimization supporting multiple outputs. By using invertible transformations (gaussian copula), the model is able to capture non-Gaussian output distributions. Experiments with artificial and real data (EEG and climate), highlight the predictive ability of the proposed model.

**Summary Of The Review:**

The proposed approach though technically simple relative to existing literature in neural and Gaussian process literature it is well motivated, technically sound and with comprehensive experimental results that support the improved predictive performance claimed by the authors.

---

> ### Author Response · Authors · 2021-11-14
> **Response by the authors of Paper3542 to Reviewer g5Qp**
>
> We would like to thank the anonymous reviewer g5Qp for their comments, and useful suggestions. We were glad to hear that the reviewer was pleased with the thoroughness of our empirical work including
> > experiments with artificial and real data (EEG and climate) [which] highlight the predictive ability of the proposed model.
>
> We were also happy that the reviewer appreciated our
> > relatively simple extension of Gaussian Neural Processes
>
> which addresses the major computational bottleneck of the FullConvGNP. We consider the simplicity of our method to be a significant merit, since the fact that practitioners can easily understand and implement it, means they are likely to adopt it in their work. As the reviewer points out, our method
> > is amenable to exact, simple and scalable optimisation supporting multiple outputs
>
> which nonetheless brings computational improvements (compared to the FullConvGNP) or performance gains (over other neural process models), which are highlighted by extensive empirical evidence.
>
> We would like to address the two suggested additions proposed by the reviewer.
>
> ### Addition 1: Estimates of computational cost and/or timing
>
> The reviewer suggested that
> > having estimates of computational cost and/or runtime experiments could be a welcome addition to the paper.
>
> **Response 1:** We would like to thank the reviewer for this very helpful suggestion. **We will add a table of measurements of runtime and memory usage for each of the models presented, before the end of the discussion period on the 22nd of November**. This should further highlight the practical benefits of our approach compared to the FullConvGNP. We would also like to provide a brief discussion of these points here. The computational and memory costs of the FullConvGNP are of order $\mathcal{O}(e^{2D})$, where $D$ is the input dimension, whereas those of ConvGNP are of order $\mathcal{O}(e^{D})$ and those of the GNP and AGNP are $\mathcal{O}(D)$ (because the size of the input layer for these scales linearly with $D$ in the latter two models). While the scaling is still exponential for the ConvGNP, it is as good as one could hope for a model using convolutions in the input space, resulting in runtimes and memory costs which are significantly lower than the FullConvGNP. This makes a great difference in practice, since applications typically involve convolutions up to $D = 3$ dimensions, which the ConvGNP can realistically handle, whereas the FullConvGNP would be too expensive to apply.
>
> ### Addition 2: Details on the lengthscale of the kernel
>
> The reviewer pointed out that
> > Something that is not discussed in the paper is the setting of the length scale of the squared exponential kernel.
>
> **Response 2:** We would like to thank the reviewer for pointing this out. In our experiments we used a fixed unit lengthscale. We do not expect that this affects the performance of the model since the neural network $g$ outputting the features which are fed to the covariance should be able to adjust their scale automatically. However, this parameter could be learnt in practice, or even outputted by the $g$ network itself. **We have added a comment clarifying this in our paper**.

---

> ### Author Response · Authors · 2021-11-22
> **Revision of manuscript**
>
> We would like to bring to the reviewer’s attention that we’ve made the promised changes to our manuscript. In particular, we have added measurements of runtime and memory costs to a new appendix (appendix B) and referenced these in the main text.

---

### Decision · Program_Chairs · 2022-01-20

**Decision:**

Accept (Poster)

**Comment:**

This paper proposes a class of neural processes that lifts the limitations of conditional neural processes (CNPs) and produces dependent/correlated outputs but that, as CNPs, is inherently scalable and it is easy to train via maximum likelihood. The proposed model is extended to multi-output regression and to capture non-Gaussian output distributions. Results are presented on synthetic data, an electroencephalogram dataset and on a climate modeling problem. The paper parameterizes the prediction map as a Gaussian, where the mean and covariance are determined using neural networks. Non-Gaussian prediction maps are obtained using copulas.

Technically speaking, the reviewers found the approach to be incremental and only marginally significant and I agree with them. Issues such as estimates of computational cost, using fixed lengthscales for the covariances and relationships/using normalizing flows have been addressed by the authors satisfactorily. Empirically, the contribution of the paper is somewhat significant, as it provides similar flexibility to other more computationally expensive processes and more general assumptions than conditional neural processes.